# BIOACOUSTIC GEOLOCATION:
# SPECIES SOUNDS AS GEOGRAPHIC SIGNALS

## ABSTRACT

Can we determine someone's geographic location solely from the sounds they hear? Are acoustic signals enough to localize within a country, state, or even city? We tackle the challenge of global-scale audio geolocation, formalizing the problem, and conducting an in-depth analysis with wildlife audio from the iNatSounds dataset. We hypothesize that bioacoustic signals contain informative geolocation cues because of well-defined geographic ranges of species. To test this, we benchmark image geolocation and soundscape mapping methods on iNatSounds. Building on these insights, we propose a hybrid approach that combines species range prediction with retrieval-based geolocation. We further ask whether geolocation improves with species-diverse recordings and spatiotemporal aggregation across neighboring samples. Finally, we extend our study to multimodal geolocation with case studies from movies that combine both audio and visual content. Our results highlight the potential of incorporating bioacoustic signals into geospatial tasks, motivating future work on species recognition and audio geolocation.

## 1 INTRODUCTION

Image geolocation has been extensively studied in the computer vision community, with methods exploring classification (Weyand et al., 2016; Vo et al., 2017; Seo et al., 2018; Berton et al., 2022; Pramanick et al., 2022; Clark et al., 2023), contrastive learning (Vivanco Cepeda et al., 2024; Klemmer et al., 2023) and multimodal representations (Zhu et al., 2021; Yang et al., 2021; Zhu et al., 2022). The best models can place 40% of images within 25 km of their actual location, rivaling top human geoguessers (Haas et al., 2024). In this work, we shift focus to predicting the recording location of audio, addressing the unique challenges of global audio geolocation.

Image geolocation relies on visual cues such as famous landmarks, landscape types, or architectural styles to infer location from a random image. By contrast, cues tied to a geographic location are less salient in audio. Early studies (Pokorny et al., 2019; Kumar et al., 2017) focus on urban sounds like jackhammers and traffic noise as signals to geolocate. We hypothesize that wildlife sounds offer another promising signal. Each species has a defined geographic range, and identifying audible species in a recording can help constrain the possible locations. Intuitively, by intersecting range maps of detected species, we can significantly narrow down the search area (Fig. 1). If the detected species have large ranges, then the search area might still involve hundreds or thousands of square kilometers. However, if any of the detected species has a restricted range, the plausible search space might be narrowed down to tens of kilometers.

Our experiments reveal that accurate geolocation is possible if we know *all* species present in a location, and performance improves as species diversity increases. However, achieving high recall in identifying audible species presents several challenges. Short audio recordings may fail to capture a representative set of vocalizing species, motivating approaches that aggregate information over broader spatial and temporal windows. Even for species that are recorded, low signal-to-noise ratios and fine-grained confusions can further limit recall. Rare species, in particular, have distinctive geographic footprints that are valuable for geolocation, yet detecting them is especially difficult. In this work, we explore these challenges in the context of bioacoustic geolocation, using iNatSounds (Chasmai et al., 2024)—a geographically and ecologically diverse wildlife audio dataset—as well as a newly curated collection of species-rich dawn chorus recordings from Xeno-Canto.

Figure 1: **Intuition for Audio Geolocation.** If a model were to recognize different species' vocalizations, the intersection of their geographic range maps could help narrow down the location. Ambient sounds like waves or airplane noises could provide additional context to refine the estimate. While our models do not explicitly detect these sound events, we hypothesize that they leverage these types of signals *implicitly* to perform geolocation.

Our work is closely related to soundscape mapping, which seeks to model the acoustic environment of a location—for example, distinguishing a busy city intersection from a rural farm. Other approaches have explored localization through intermediate modalities, such as images paired with audio (Sastry et al., 2025), but these too do not directly solve the audio geolocation problem. We argue that audio geolocation presents a unique set of challenges, provide strong baselines that demonstrate feasibility of the task in bioacoustic domains and establish a foundation for future work.

Audio geolocation has numerous impactful potential applications. Coarse geographic context inferred from audio can enable apps such as Merlin Sound ID to constrain the set of plausible species in areas without cell coverage or GPS availability. Soundscapes inherently capture key aspects of a habitat, and shifts in predicted locations of a region over time can be used to monitor ecological change without expensive species labeling. Beyond ecology, audio geolocation can also be applied for digital forensics (Sec. 4.5), search and rescue (localizing distress calls), and data privacy (understanding and obfuscating prominent location cues). We see our study as laying the groundwork for future efforts and a first step towards ever more diverse audio geolocation applications.

We summarize our main contributions as follows:

1. To the best of our knowledge, this is the first study on global-scale bioacoustic geolocation. We formalize the problem, benchmark image-based and soundscape mapping methods on iNat-Sounds (Chasmai et al., 2024), and propose a novel approach that uses species range predictions to improve geolocation.

2. We investigate the potential of species information by constructing species oracles based on range estimation models. To study more realistic settings, we also introduce XCDC, a new dataset of species-rich dawn chorus recordings curated from Xeno-Canto.

3. We demonstrate the benefits of spatiotemporal aggregation for capturing richer species information and overcoming the limitations of short audio recordings.

4. We introduce multimodal geo-forensics through case studies of movie audio and imagery, illustrating potential real-world applications of multimodal geolocation.

## 2 RELATED WORK

**Audio Geolocation.** The task of explicit geolocation of audio recordings remains largely under-explored. Several early studies (Pokorny et al., 2019; Kumar et al., 2017; Choi et al., 2015; Friedland et al., 2010; 2011) investigate audio and multimodal geolocation, but are limited to distinguishing sounds within or among specific regions. Motivated by the growing interest in image geolocation, we revisit and scale up the problem of audio geolocation from a modern perspective, aiming to address it at a global scale. Insights gained from our explorations in natural domains could also contribute to broader advancements in audio geolocation, as wildlife sounds such as birds and insects are also prevalent in urban settings.

**Image Geolocation.** Initial methods for image geolocation approached the problem through the lens of retrieval. To geolocate a ground image, they retrieved similar geo-tagged ground images (Hays & Efros, 2008) or cross-view satellite images (Workman et al., 2015; Liu & Li, 2019;

Shi et al., 2020; Zhu et al., 2021; Yang et al., 2021; Zhu et al., 2022). Focus gradually shifted to feed-forward classification models that could directly predict location from an image by splitting the surface of the earth into several geo-cells and predicting the correct geo-cell (Berton et al., 2022; Weyand et al., 2016; Vo et al., 2017; Muller-Budack et al., 2018; Clark et al., 2023; Pramanick et al., 2022; Seo et al., 2018). To overcome grid resolution constraints, some works explore adaptive grids (Seo et al., 2018) and hierarchical approaches (Clark et al., 2023). Haas et al. (2024) incorporate political and administrative boundaries while creating their grids and beat one of the world's foremost professional GeoGuessr players (GeoGuessr, 2013).

Following the release of CLIP (Radford et al., 2021), recent methods (Klemmer et al., 2023; Vivanco Cepeda et al., 2024) have revisited the retrieval approach. These methods treat the location itself as a modality and learn location embeddings aligned with images. GeoCLIP (Vivanco Cepeda et al., 2024) learns a location encoder that utilizes random Fourier features to capture high frequency details and learn better location features. SatCLIP (Klemmer et al., 2023) encodes location via spherical harmonics and learns a joint embedding space with paired satellite images. Zhou et al. (2024) propose to leverage LLMs (Achiam et al., 2023; Touvron et al., 2023) and incorporate domain knowledge with Retrieval Augmented Generation (RAG) (Cai et al., 2022). In this work, we shift the focus from visual to the acoustic modality. Audio presents a unique challenge, as it typically carries less explicit and globally consistent geographic information than visual data. We extend and benchmark multiple image geolocation methods across popular paradigms in an effort to establish a comprehensive benchmark for the novel task of bioacoustic geolocation.

**Soundscape Mapping and Audio-Location Alignment.** In the acoustic space, researchers are also interested in the complementary problem of soundscape mapping. Instead of identifying geographically unique signals in audio recordings, the focus is to learn common patterns in sounds from a particular location. The task is posed as a retrieval of audio conditioned on location. Prior research focus on understanding the soundscapes of either a few cities (Aiello et al., 2016) or for the entire earth (Salem et al., 2018). GeoCLAP (Khanal et al., 2023) and PSM (Khanal et al., 2024) use satellite images to represent geographic location and learn a shared embedding space between overhead images and ground audio. The domain of wildlife sounds also provides a unique set of challenges not as prominent in urban settings. Taxabind (Sastry et al., 2025) learns a shared feature space for 6 modalities related to species and demonstrate its benefit to ecological problems. However, they too do not focus on audio geolocation and have *implicit* audio-location pairing with ground images as an intermediary. While these methods can be implicitly used for geolocation, the unique challenges of global audio geolocation remain largely underexplored, a critical gap this work aims to fill.

**Related Audio Tasks.** In acoustic analysis, prior work also focus on localizing sources within specific scenes relative to the microphone position (Sun et al., 2017; Liu et al., 2019; Wu et al., 2021; Chung et al., 2022; Zhang et al., 2018; Dang et al., 2019; Perotin et al., 2019; Chen et al., 2020; 2021; 2023). While the structure of this problem is very similar to our study of audio geolocation, the two tasks require very different signals: features learned for one may not be predictive of the other. There has also been some prior work in geolocating speech, either as a proxy task for identifying accents and dialects (Van Leeuwen & Orr, 2016; Lohfink, 2017; Dehak et al., 2010; Foley et al., 2024), or to improve speech recognition (ASR) (Xiao et al., 2018; Bell et al., 2015). Our work focuses on the location of the location of the recording environment instead of the origin of a subject, for which human speech often provides insufficient or ambiguous geographic information.

## 3 PROBLEM SETUP AND METHODOLOGY

In this work, we conduct experiments using iNatSounds (Chasmai et al., 2024), a dataset of 230K audio recordings, with an average duration of around 20s. Each recording is annotated with a species, recording date, and geographic location. The training split includes audio from over 5,500 species and spans diverse regions across the Americas, Africa, Europe, Asia, and Australia. However, the distribution is imbalanced, with a bias toward major population centers. See supplementary Fig. A4 for the geographic distribution of iNatSounds and Fig. 2 for some examples from around the world.

**Preliminaries.** Given an audio recording $\mathbf{x}$, our goal is to predict the coordinates $\mathbf{y} = (\text{latitude}, \text{longitude})$ of the location where $\mathbf{x}$ was recorded. We evaluate the geolocation error by computing the Haversine distance (Gade, 2010) between the predicted and true coordinates. Given a dataset $\mathcal{D} = \{(\mathbf{x}_i, \mathbf{y}_i)\}_{i=1}^{N}$ of audio recordings $\mathbf{x}_i \in \mathcal{X}$ and ground truth locations $\mathbf{y}_i$, we mea-

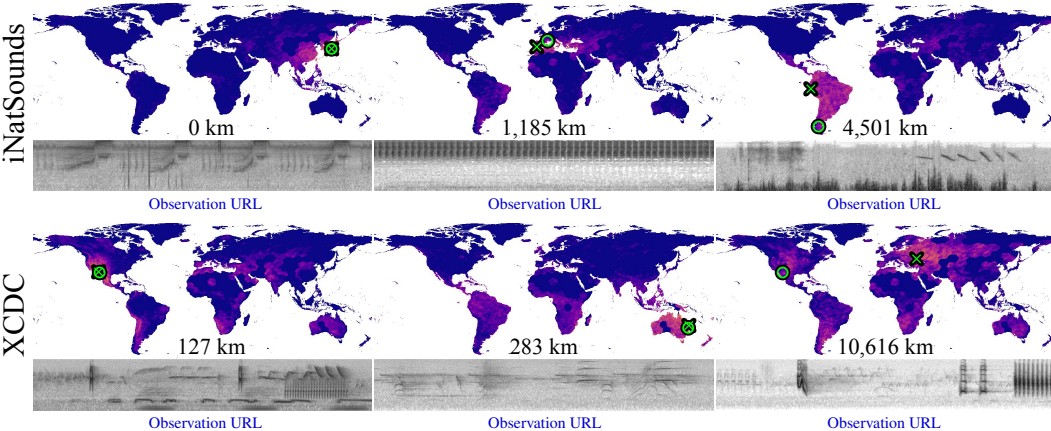

Figure 2: **Geolocation Predictions.** Sample predictions of our method on iNatSounds (Chasmai et al., 2024) and XCDC. Heatmap shows the unnormalised likelihood of each location. Green crosses denote the final prediction (argmax) and green circles denote true location. Geolocation error reported below each heatmap. URLs to the original observations (clickable) provided below each spectrogram. Harshness of the region threshold (200km) can be better appreciated through the bottom left and center exemplars. We present additional visualizations in supplemental Fig. A2, A3.

sure the performance of a method using the median geolocation error across $\mathcal{D}$. Following standard practice in image geolocation (Weyand et al., 2016; Vivanco Cepeda et al., 2024; Haas et al., 2024), we also report "Percentage at Threshold," where a prediction is considered correct if the geolocation error is within a specific threshold. We use thresholds of 25 km, 200 km, 750 km, and 2,500 km, corresponding to city, regional, country, and continental scales. Refer to Fig. 2 and supplementary Fig. A1 for illustrations of these scales.

We aim to train a geolocation model $g : \mathcal{X} \to \mathcal{L}$ where $\mathcal{L}$ is the space of all possible locations. This model is decomposed as $g = h_\phi \circ f_\theta$, where $f_\theta : \mathcal{X} \to \mathbb{R}^d$, is an audio encoder with parameters $\theta$, and $h_\phi : \mathbb{R}^d \to \mathcal{L}$ is a decoder with parameters $\phi$ responsible for geolocating an audio feature. The location can be encoded in different ways, and the definition of $\mathcal{L}$ changes accordingly. It can be simply (lat, lon) (for regression), a geographic bin label (for classification), or a learned embedding (for retrieval). Although the location encoding does not need to be reversible, each prediction in $\mathcal{L}$ should be mappable to a corresponding (lat, lon) for evaluation.

**Audio Encoder.** We adopt a vision-inspired approach to acoustic analysis by converting audio to a spectrogram "image" and extracting features with computer vision backbones. Model architecture and pre-training strategies are described in Sec. 4.1 and spectrogram settings in supplemental Sec. A.5. We ablate the choice of audio encoders in Sec. 4.3.

**Retrieval for Geolocation.** We pose the problem of audio geolocation as retrieval, where $h_\phi$ is responsible for computing the similarity of $f_\theta(\mathbf{x})$ (query) with a predefined collection of embeddings (keys). Recent prior work (Vivanco Cepeda et al., 2024) has explored location encoders $l_\psi : \mathbb{R}^2 \to \mathbb{R}^d$, that learn dense representations for two-dimensional (lat, lon) values. We train $h_\phi$ to project audio features into the embedding space of a location encoder. Retrieval can then be done at test time by comparing the predicted embedding $g(\mathbf{x}) \in \mathcal{L}$ against a gallery of precomputed location embeddings, and returning the coordinates of the most similar location embedding. We use the training set itself to construct our gallery, so the predicted coordinates $\hat{\mathbf{y}}$ can be expressed as $\hat{\mathbf{y}} = \arg\max_{\mathbf{y}_j \in \mathcal{D}^{Train}} g(\mathbf{x})^\top l_\psi(\mathbf{y}_j)$.

**Species Oracles.** We briefly diverge to introduce a diagnostic tool designed to probe the limits of bioacoustic geolocation. We hypothesize that presence and absence of species is a strong signal for geolocation. One immediate challenge for testing this hypothesis is the unrealistic expectation of capturing sounds from all species in a location. In theory, if a recordist were to continuously capture audio at a location over an entire year, we might expect to record most species that occur there. With this complete "checklist" of species, how accurately can we predict recording location?

While year-long recordings are impractical, we can simulate these comprehensive checklists using precomputed species geographic range maps. For a given location on Earth, we determine which

species' ranges contain that location and construct a checklist of species that are present or absent. Although expert-verified range maps are not available for all species, recent advancements (Cole et al., 2023; Hamilton et al., 2024) have enabled joint prediction of range maps for tens of thousands of species. We leverage one such model (SINR (Cole et al., 2023)), to construct range maps for all species in the iNatSounds dataset. Finally, we learn $h_\phi$ to perform retrieval style geolocation using these checklists instead of audio features. Note that checklist construction assumes oracle access to the true location. These experiments do not reflect a realistic bioacoustic geolocation setting, but instead serve as upper bounds for species-based approaches.

**AG-CLIP.** Our proposed method, AudioGeo-CLIP, builds on top of GeoCLIP (Vivanco Cepeda et al., 2024) and includes an auxiliary task of predicting SINR (Cole et al., 2023) checklists from audio. Concretely, we add a checklist decoder and a BCE loss with the species checklists in addition to the usual location decoder $h_\phi$ and its contrastive loss. While a particular recording will likely capture only a small fraction of species whose ranges overlap the recording location, predicting the species checklist can encourage the encoder to attend not only to foreground vocalizations but also to background ambient sounds, learning to associate them with species likely to be encountered in that habitat. See supplemental Fig. A6 for a schematic.

**Handling Variable Length.** For each recording, we construct a collection of 3s "clips" using a sliding window. Each clip is geolocated independently and then the average of all clip-predictions is used as the final prediction for that recording. We experiment with alternate poolings in Sec. 4.3.

## 4 EXPERIMENTS

### 4.1 IMPLEMENTATION DETAILS

In the following experiments, the audio encoder $f_\theta$ is a trainable MobileNet-V3 (Howard et al., 2019) model pretrained for species classification on the iNatSounds dataset. These models expect spectrograms that have been resized to 224x224x3. For AG-CLIP, $h_\phi$ is composed of two 2-layer MLPs, each with hidden dimensions of 128, responsible for projecting features to GeoCLIP embeddings and SINR checklists, respectively. In Sec. 4.3, we further explore the impact of different network architectures and pretraining strategies for both audio and location encodings. All models are trained using SGD with Nesterov acceleration. We use the validation set of iNatSounds to select the learning rate and other hyperparameters. See Supp. Sec. A.5 for detailed model configurations.

### 4.2 PERFORMANCE ON iNATSOUNDS

We present the performance of geolocation methods on the iNatSounds test set in Table 1.

**Naive Baseline.** We start with a naive baseline that samples a random location from the training distribution for each test recording. This baseline yields a regional accuracy of just 1.1%, suggesting that the dataset is sufficiently diverse and not dominated by a few geographic hotspots.

**Species Oracles.** To contextualize subsequent results, we first present upper-bound performance using species checklists constructed with oracle access to the true location. We binarize SINR (Cole et al., 2023) model outputs using a threshold of 0.1 to create binary vectors, or "checklists", indicating species presence and absence at each location. We then train a linear model to serve as $h_\phi$, predicting GeoCLIP embeddings and geolocating via retrieval. Our results indicate that with a *full checklist*, almost perfect region level performance is possible, confirming our hypothesis that species information serves as a strong geolocation signal. However, knowing all species that may be found at a location is not realistic with reasonably sized audio recordings and noisy or incomplete species identification. To simulate imperfect recall, we omit a random subset of species from the checklist. With *50% corruption*, regional accuracy remains relatively high at 83.3%. If only *10 randomly selected species* are retained, region performance drops to a still strong 37.6%.

**Regress.** Regression methods tend to do the worst overall, with region level performances of 2.9% and 4.2% for Euclidean and Haversine, respectively. Haversine distance more accurately captures geolocation error, which is the likely cause of its better performance as a loss function.

**Classify.** For classification, we use the H3 library (Brodsky, 2018) to divide the world into a hexagonal grid. $h_\phi$ is implemented as an N-way classifier where N is the number of hexagons in the

Table 1: **Geolocation on iNatSounds**. Experiments on iNatSounds test set. The naive random predictor and species oracles contextualize performance. We train our models to geolocate using regression, classification, explicit sampling from range maps and retrieval. Geolocation performance is evaluated by median error (km) and accuracies (%) at different distance thresholds. We report mean $\pm\ std$ from 3 runs. †: Off the shelf models.

| | Experiment | ↓ Median Error (km) | ↑ City 25km | Region 200km | Country 750km | Continent 2500km |
|---|---|---|---|---|---|---|
| Naive | Rnd Train Loc | $7475 \pm 32$ | $00.1 \pm 0.0$ | $01.1 \pm 0.0$ | $06.6 \pm 0.1$ | $22.8 \pm 0.3$ |
| Species Oracles | Full checklist | $15 \pm 00$ | $64.0 \pm 0.4$ | $97.8 \pm 0.0$ | $99.9 \pm 0.0$ | $100. \pm 0.0$ |
| | 50% Corrupted | $54 \pm 00$ | $32.8 \pm 0.0$ | $83.3 \pm 0.1$ | $98.8 \pm 0.0$ | $100. \pm 0.0$ |
| | Keep random 10 | $325 \pm 02$ | $10.0 \pm 0.1$ | $37.6 \pm 0.2$ | $76.6 \pm 0.2$ | $98.3 \pm 0.0$ |
| Regress | Euclidean | $1884 \pm 05$ | $00.1 \pm 0.0$ | $02.9 \pm 0.0$ | $23.4 \pm 0.2$ | $57.9 \pm 0.1$ |
| | Haversine | $1602 \pm 13$ | $00.1 \pm 0.0$ | $04.2 \pm 0.2$ | $27.7 \pm 0.3$ | $61.8 \pm 0.2$ |
| Classify | Res-0 ($430 \times 10^4 \mathrm{km}^2$) | $1323 \pm 09$ | $00.0 \pm 0.0$ | $01.2 \pm 0.0$ | $22.6 \pm 0.0$ | $68.7 \pm 0.2$ |
| | Res-2 ($8.6 \times 10^4 \mathrm{km}^2$) | $1326 \pm 12$ | $00.3 \pm 0.0$ | $16.3 \pm 0.2$ | $36.5 \pm 0.4$ | $64.1 \pm 0.2$ |
| | Hierarchical ($0 \rightarrow 1 \rightarrow 2$) | $1117 \pm 06$ | $00.3 \pm 0.0$ | $16.6 \pm 0.2$ | $40.0 \pm 0.1$ | $68.7 \pm 0.2$ |
| Species Ranges | Annotated Species | $1263 \pm 01$ | $00.3 \pm 0.0$ | $10.8 \pm 0.1$ | $35.7 \pm 0.1$ | $72.3 \pm 0.1$ |
| | Predicted Sp (Top 1) | $1664 \pm 03$ | $00.2 \pm 0.0$ | $08.2 \pm 0.1$ | $28.7 \pm 0.0$ | $62.5 \pm 0.1$ |
| | Predicted Sp (All) | $1113 \pm 02$ | $00.3 \pm 0.0$ | $09.3 \pm 0.1$ | $37.7 \pm 0.1$ | $\mathbf{72.7} \pm 0.1$ |
| Retrieve | GeoCLAP† | $6856 \pm 00$ | $00.2 \pm 0.0$ | $01.3 \pm 0.0$ | $07.0 \pm 0.0$ | $24.9 \pm 0.0$ |
| | Taxabind† | $4944 \pm 00$ | $00.4 \pm 0.0$ | $02.2 \pm 0.0$ | $11.9 \pm 0.0$ | $35.3 \pm 0.0$ |
| | AG-CLIP (**ours**) | $\mathbf{1082} \pm 11$ | $\mathbf{06.4} \pm 0.1$ | $\mathbf{17.2} \pm 0.1$ | $\mathbf{41.0} \pm 0.3$ | $71.2 \pm 0.2$ |

grid. We experiment with two grid resolutions, defined by the area of each cell. More details and visualization of these grids are presented in Supp. Sec. A.3. Lower resolution (bigger cells) leads to better continental performance while higher resolution leads to better regional and country level performance. A hierarchical approach gets the best of both, performing similar to low resolution for continent (68.7% vs 68.7%) and better than high resolution for country (40.0% vs 36.5%).

**Species Ranges.** We next evaluate performance when explicit species information is available. Each iNatSounds recording includes a species annotation. By sampling the species' SINR (Cole et al., 2023) likelihood distribution over a geographic grid, we can geolocate the recording. Note that SINR is trained on a much larger corpus of species occurrence data, which may offer an advantage over other methods. Using the annotated species, we see a region accuracy of 10.8% and a continent accuracy of 72.3%. However, the availability of ground-truth target species annotations at test time is unrealistic; instead, we can train a classifier to predict the target species from the audio. Using the SINR distribution of the top-1 predicted target species drops the region and continent performance to 8.2% and 62.5% respectively. Rather than using a single predicted species for each recording, we can use the per-species scores given by our species classifier to weigh and combine SINR likelihood maps for all species. This weighted combination of species likelihood maps is slightly better than even the *annotated* target species (country 37.7% vs 35.7%), which may be an artifact of these classifiers' ability to capture background species to some extent (Chasmai et al., 2024).

**Retrieve.** For retrieval, we explore methods that jointly learn audio and location encoders. Geo-CLAP (Khanal et al., 2023) and Taxabind (Sastry et al., 2025) are soundscape mapping methods that have their own audio and location encoders, which we use off-the-shelf without any additional training. GeoCLAP performs poorly, likely due to domain shift, as it was trained on urban sounds from SoundingEarth (Heidler et al., 2023). On the other hand, Taxabind, which was trained on a different subset of iNaturalist (iNaturalist), achieves better results with a country level accuracy of 11.9%. However, it still underperforms relative to our models, likely because our models benefit from explicitly paired audio and location data, whereas Taxabind relies on only the implicit pairing available through images. Our method, AG-CLIP, achieves the best performance overall, with region and country level performances of 17.2% and 41.0%, respectively.

**Performance at Different Distance Thresholds.** Fig. 3 (left) shows CDF curves for various methods, plotting the fraction of test recordings (y-axis) correctly geolocated within a given distance (x-axis). These curves reveal how models perform across geospatial scales. Geolocation based on species checklists, even with 50% corruption, is best across scales. Among audio geolocation

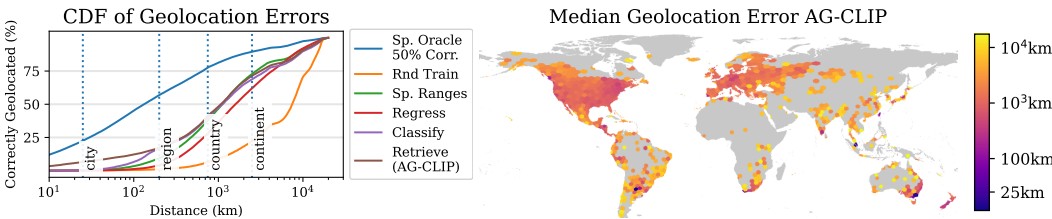

Figure 3: **Geolocation Error Trends.** Left: Cumulative distribution of geolocation errors for AG-CLIP and baseline models. Table 1 accuracy metrics correspond to points on these curves. Right: Median of AG-CLIP geolocation errors binned by H3 (Brodsky, 2018) grid cells, capturing spatial variation in performance.

| Model | Pre | FT | Reg. | Cont. |
|---|---|---|---|---|
| Wav2CLIP | VGG | ✗ | 03.8 | 30.3 |
| CLAP | LAION | ✗ | 05.1 | 34.6 |
| MobV3 | ImgNet | ✓ | 14.4 | 64.9 |
| MobV3 | iNat | ✓ | **17.2** | **71.2** |

(a) Audio Encoders

| Experiment | City | Reg. | Cont. |
|---|---|---|---|
| SatCLIP | 1.4 | 14.7 | 63.2 |
| SINR | 2.2 | 17.1 | 68.6 |
| GeoCLIP | **6.4** | 17.0 | 68.5 |
| ” + checklist | **6.4** | **17.2** | **71.2** |

(b) Location Encoders

| Experiment | Reg. | Cont. |
|---|---|---|
| Average | **17.2** | 71.2 |
| Max Pool | 16.5 | 68.6 |
| Cluster | 15.5 | 67.9 |
| Transformer | 16.8 | **71.5** |

(c) Variable Length

Table 2: **Ablations.** Importance and alternative choices of different components of AG-CLIP. We include Region (Reg. 200km) and Continent (Cont. 2500km) performance on iNatSounds test set.

methods, retrieval based AG-CLIP does best at finer scales like the city and region. Hierarchical classification shows a jump in performance around the region level and is close to AG-CLIP for coarser scales. This may be due to the finest grid cells being similar in size to a region, limiting the model's ability to predict at finer resolutions. Explicit species identification with range analysis is relatively poor for finer scales, but overtakes AG-CLIP around the 1500 km mark.

**Spatial Variation.** Fig. 3 (right) shows how geolocation performance of AG-CLIP varies geospatially. We associate each recording with a resolution 2 hexagon from H3 (Brodsky, 2018) and compute the median error per hexagon. Darker (bluer) regions indicate better geolocation. The model performs better in regions with more training data, like the US and Europe. Interestingly, certain areas such as Central America, South Africa, Taiwan, Eastern Australia, and New Zealand show particularly good performance, possibly hinting at the presence of distinctive soundscapes here.

### 4.3 ABLATIONS

We ablate the audio encoders, location encoders, and temporal aggregation strategies in Table 2. Off-the-shelf audio encoders like Wav2CLIP (Wu et al., 2022) and CLAP (Elizalde et al., 2023) perform worse than pretraining on iNatSounds (Table 2a). Starting with ImageNet pretrained weights instead of iNatSounds also reduces performance, with 7% and 3% drops at the continent and region levels.

Next, we ablate the location encoder in Table 2b. With SatCLIP (Klemmer et al., 2023), we observe consistently worse performance than SINR (Cole et al., 2023) or GeoCLIP (Vivanco Cepeda et al., 2024). SINR is comparable to GeoCLIP at the region and continent levels, but tapers off at the city level. Better performance of GeoCLIP may be because of its alignment with ground-level imagery, unlike SatCLIP (satellite images) or SINR (species observations). The auxiliary checklist loss introduced in AG-CLIP improves performance by 2.7% at the continent level, though the gains diminish at finer scales.

Finally, we explore alternate pooling methods to handle variable length audio in Table 2c. Our default is a simple average. We see a drop in performance if max-pooling is used instead. We could also first cluster the embeddings via K Means (k=5) and then use the centroid of the largest cluster as the aggregate. While better than max-pooling, this is still worse than averaging. We also try training a transformer model to pool frozen embeddings (see supplemental Sec. A.6 for more details). The transformer performs well at the continent level, but we prefer averaging because of its simplicity and lack of additional training.

Table 3: **Species-Rich Audio with XCDC**. Left: Models trained on iNatSounds training set and evaluated on XCDC. We report mean of 3 runs. Right: Geolocation with species ranges for randomly sampled subsets of ground truth species. We plot mean, std and range (min-max) over 100 runs.

| Experiment | Error | City | Region | Cou. | Cont. |
|---|---|---|---|---|---|
| Species Ranges (True) | 449 | 00.5 | 25.8 | 68.8 | 99.0 |
| Species Ranges (Predicted) | 1097 | 00.1 | 02.5 | 27.6 | 80.0 |
| Classification (Hierarchical) | 1116 | 00.0 | 05.3 | 31.7 | 69.7 |
| AG-CLIP (**ours**) | 1112 | 00.2 | 04.3 | 26.3 | 71.9 |

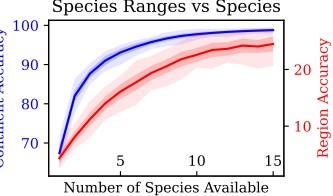

Table 4: **Spatiotemporal Aggregation**. Left: AG-CLIP geolocation can be improved by grouping recordings from the same neighborhood, over a year, month or week. Right: Distributions of unique species per collection.

| Experiment | Error | City | Region | Cou. | Cont. |
|---|---|---|---|---|---|
| No Grouping | 1082 | 06.4 | 17.2 | 41.0 | 71.2 |
| Resolution-6 (36 km$^2$) + Week | 890 | 07.9 | 19.9 | 45.8 | 76.5 |
| Resolution-6 (36 km$^2$) + Month | 802 | 08.5 | 21.2 | 48.2 | 78.3 |
| Resolution-6 (36 km$^2$) + Year | 651 | 10.8 | 25.9 | 55.6 | 84.0 |
| Resolution-5 (253km$^2$) + Year | 520 | 13.2 | 30.4 | 62.1 | 88.2 |

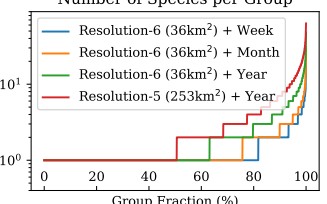

## 4.4 GEOLOCATING SPECIES-RICH AND SPATIOTEMPORALLY AGGREGATED RECORDINGS

Our species oracle experiments (Table 1) demonstrate that geolocation accuracy increases with more knowledge of the species present at the recording location. However, short recordings often capture only a few species, limiting this important signal. To evaluate whether increased species diversity improves geolocation, we explore two strategies: (1) sampling long, species-rich recordings from the dawn chorus and (2) aggregating short recordings from the same spatiotemporal neighborhood.

**Species-Rich Audio.** We construct a dataset of species-rich audio by sampling dawn chorus recordings from Xeno-Canto, a global archive of natural sound recordings. The dawn chorus is a period of intense, multi-species vocal activity that occurs near sunrise, particularly during the breeding season (Gil & Llusia, 2020; Weldy et al., 2024). To isolate these species-rich recordings, we select audio that (1) was recorded during the spring dawn chorus, (2) is at least 3 minutes in duration, and (3) contains annotations for at least 10 distinct species. This filtering yields 576 recordings with associated geographic coordinates and species labels. We refer to this as Xeno-Canto Dawn Chorus (XCDC), a new benchmark for evaluating geolocation from species-rich soundscapes.

Our results are shown in Table 3. We reconfirm the value of species information: geolocation accuracy increases as more ground truth species are provided (Table 3, right), with full species knowledge yielding a region-level accuracy of 25.8% (Table 3, top row). However, model-based approaches that rely on predicted species ranges, hierarchical classification, or retrieval achieve substantially lower performance ($\leq 5.3\%$), roughly equivalent to knowing only a single species. We observe similar trends for WABAD (Pérez-Granados et al., 2025), an in-the-wild, passive acoustic monitoring dataset (See supplemental Sec. A.9). This suggests that current models struggle to exploit the multi-species signals present in these complex soundscapes. One likely explanation is a distribution shift: most iNatSounds recordings contain fewer species, and our models may not generalize well to audio with significant species overlap. This finding is reinforced by our spatiotemporal aggregation experiments, where recordings with higher species diversity and low species overlap yield better results.

**Spatiotemporal Aggregation.** We group iNatSounds test recordings by location and time to simulate spatiotemporal aggregation. Spatial grouping is performed using H3 hexagons at resolutions 5 and 6, while temporal grouping spans the full year, individual months, or individual weeks. To geolocate a group, we first apply our model independently to each recording, then average the predicted location distributions across the group. The aggregated prediction is assigned to all recordings in the group, allowing direct comparison with recording-level results from Table 1.

Table 4 reports AG-CLIP performance under different spatiotemporal aggregation strategies. When aggregating recordings within 36 km$^2$ neighborhoods and a week, the region performance improves

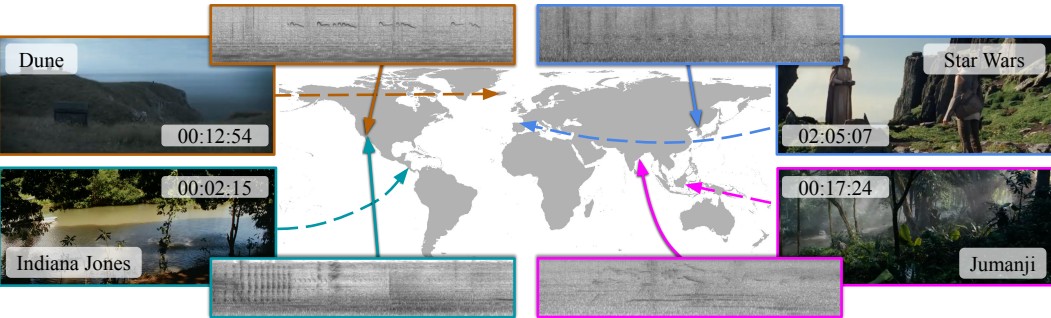

Figure 4: **Multimodal Geo-Forensics with Movie Clips.** Geolocation predictions for audio and video frames from scenes in four movies. Arrows on the map indicate the predicted locations for each modality. Discrepancies between modalities reveal potential artifacts introduced during post-production.

to 19.9%. This modest improvement reflects a limited aggregation: as shown in the right panel of Table 4, over 80% of groups in the weekly grouping contain only a single species. When aggregation is extended to a month, performance improves to 21.2%. Aggregating recordings across the full year within $36\,\mathrm{km}^2$ and $253\,\mathrm{km}^2$ neighborhoods improves region-level accuracy from 17.2% (no grouping) to 25.9% and 30.4%, respectively. These gains relative to the ungrouped baseline suggest that the model is able to leverage complementary species information spread across multiple recordings.

### 4.5 MULTIMODAL GEO-FORENSICS

Live broadcasts of events, such as professional golf tournaments, often enhance their ambiance by inserting bird vocalizations. However, observers have noted cases where producers mistakenly included species not found at the event location (slate.com). Similar mismatches arise in film, where audio effects may not correspond to the geographic setting depicted on screen (reddit; ornitheology.com). These inconsistencies offer a unique opportunity for *multimodal geo-forensics*.

In Fig. 4, we analyze scenes from four Hollywood films that had been previously flagged by viewers for mismatches between visual setting and soundscape. In all cases, audio- and image-based geolocation produced divergent predictions, confirming the presence of modality inconsistencies. Jumanji showed the closest alignment between modalities at 4480 km, while Star Wars exhibited the largest divergence at 9092 km. These qualitative results highlight the potential of multimodal geo-forensics as a novel application domain, leveraging cross-modal cues to identify geographic inconsistencies. We view this as a promising direction for future work at the intersection of vision, sound, and place. See supplemental Sec. A.10 for more details.

## 5 CONCLUSION

Geolocating arbitrary audio is challenging, but recordings containing bioacoustic signals offer promising geographic cues. Wildlife sounds can be explicitly identified and linked to known species ranges, or implicitly leveraged by end-to-end models. Our species oracle experiments show that when at least 10 species are detected, geolocation within 200 km is possible 37% of the time (Table 1). However, in species-rich recordings from XCDC, our species classification models achieve only 2.5% region accuracy (Table 3), highlighting a gap in current multi-species detection capabilities. On shorter, focused recordings, species-range models perform better (9.3%), and end-to-end models achieve 17.2% region accuracy (Table 1).

Future work could improve species classification in complex soundscapes, develop architectures that better capture long-range acoustic context, and explore structured fusion across multiple recordings. Collecting training data with dense species sounds may help address distributional shifts relative to datasets like XCDC. Better sampling or reweighting strategies could help alleviate the effects of geographical biases in iNatSounds. Our multimodal geo-forensics case studies highlight opportunities for combining audio and visual signals, opening new avenues for future exploration. This work lays the foundation and motivates future research in the challenging problem of bioacoustic geolocation.

## ETHICS STATEMENT

**Data Usage and Licensing.** The observations in iNatSounds are licensed for research use (Chasmai et al., 2024). We do not modify or re-release any data from iNatSounds. For XCDC, we collect recordings from openly available data in XenoCanto. As per their website, the usage of each recording can be specified by the user who uploaded it. From the recordings in our filtering criteria, we pick the most restrictive one (CC BY-NC-ND), and release XCDC with this license. For the recordings that we present as prediction visualizations (Fig 2, A2 and A3), we link the original observations and credit the users who contributed those recordings.

Human voice or other personally identifiable content can be present in iNatSounds and XCDC. Since the recordings are openly available and licensed for research use, we do not obfuscate or modify the data for anonymization.

**Privacy Concerns.** An adversary with the ability to infer recording locations of any online content just from the audio can violate the creator's privacy. On the other hand, by understanding the geographic cues embedded in audio, it may be possible to develop methods to suppress or obfuscate them, thereby mitigating such risks. At its current stage, our work focuses on natural sounds, and even our best models can get the city correct only 6% of the time. This suggests that the immediate privacy risks are somewhat limited. That said, we recognize that as audio geolocation models improve, these risks will grow and become an important consideration for the field.

**Geographical Biases.** iNatSounds exhibits geographic bias towards Western countries, particularly in the Northern Hemisphere. Our models exhibit better geolocation performance in regions with greater training data coverage (Fig. 3). As the evaluation split of iNatSounds exhibits similar biases, the reported performance may be overestimated.

**LLM Usage.** We used ChatGPT to aid and polish paper writing. We primarily used the LLM to paraphrase individual sentences or short paragraphs to correct grammar and improve overall flow.

## REPRODUCIBILITY STATEMENT

Upon acceptance, we will release all code and detailed instructions for benchmarking geolocation models as well as for training our method and baselines. iNatSounds (Chasmai et al., 2024) is already publicly available and we will release XCDC with appropriate licensing and links to original Xeno-Canto observations.

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

# A APPENDIX

We start with a visualizatoin of the distance thresholds used by our evaluation metrics in Sec. A.1. We follow this with additional prediction visualizations in Sec. A.2. Next, we visualize the different ways to represent location for classification (Sec. A.3) and retrieval (Sec. A.4). We next describe additional implementation details left out in the main paper (see Sec. A.5). In particular, we elaborate on the methodology with a block diagram, describe the spectrogram creation process, and report tuned hyperparameters and other configurations. We also share additional details for GeoCLAP and TaxaBind as well as the transformer pooling ablation (Sec. A.6) in Table 2 (main paper). Next, in Sec. A.7, we report additional ablation on Location Galleries. Additional results and standard deviations for XCDC are reported and discussed in Sec.A.8. We also experiment with an in-the-wild passive acoustic monitoring dataset in Sec. A.9. Next, we report additional details for geoforensics experiments (Sec. A.10) and present additional experiments on multimodal geolocation (Sec. A.11), species affinities (Sec. A.12) and multimodal retrieval (Sec. A.13). We conclude with a discussion on compute resources in Sec. A.14.

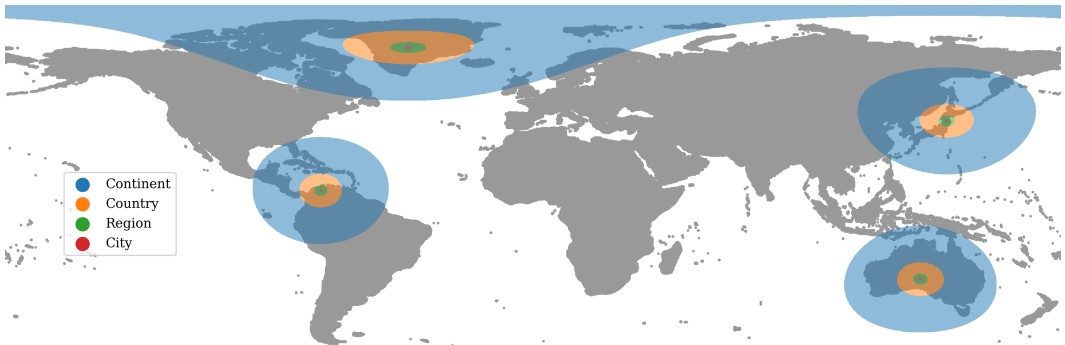

Figure A1: **Scales of Geolocation.** We plot all points within different thresholds used for calculating our geolocation metrics. These are not circles because we use haversine distance. We show these scales centered at a few different locations since the size changes at different points on earth.

## A.1 SCALES OF GEOLOCATION

For better understanding of the distance thresholds used in evaluation and an appreciation of the difficulty of this task, please see their visualization in Fig A1. A prediction is considered correct at a given level if it falls within the corresponding area centered on the ground-truth location. Note the great difference in scale between finer levels like the city (barely bigger than a dot) and coarser levels like the country and continent. While 200km for the region level may sound large, the small green areas in Fig A1 show how small it is compared to the size of the world. Even the continent threshold is actually harsher than it sounds, and covers only about the size of Australia.

## A.2 MORE PREDICTION VISUALIZATIONS

See additional prediction visualizations of AG-CLIP on iNatSounds and XCDC in Fig A2 and Fig A3 respectively. These are a mix of success and failure cases of the model.

## A.3 CLASSIFICATION GRIDS.

We visualize the different resolution hexagonal grids we use for classification in Fig A4. We train the model to predict a hexagon from the list of all possible hexagons at a particular resolution. At test time, we use the center of the predicted hexagon as the geolocation prediction. As we go to finer resolutions, the ability of a classifier to pinpoint to finer scales increases. At the same time, the total number of hexagons also increases, which reduces classification performance.

We also show the distribution of iNatSounds training set on the same plots. The distribution is highly imbalanced, with North America and Europe having significantly higher number of recordings than other areas.

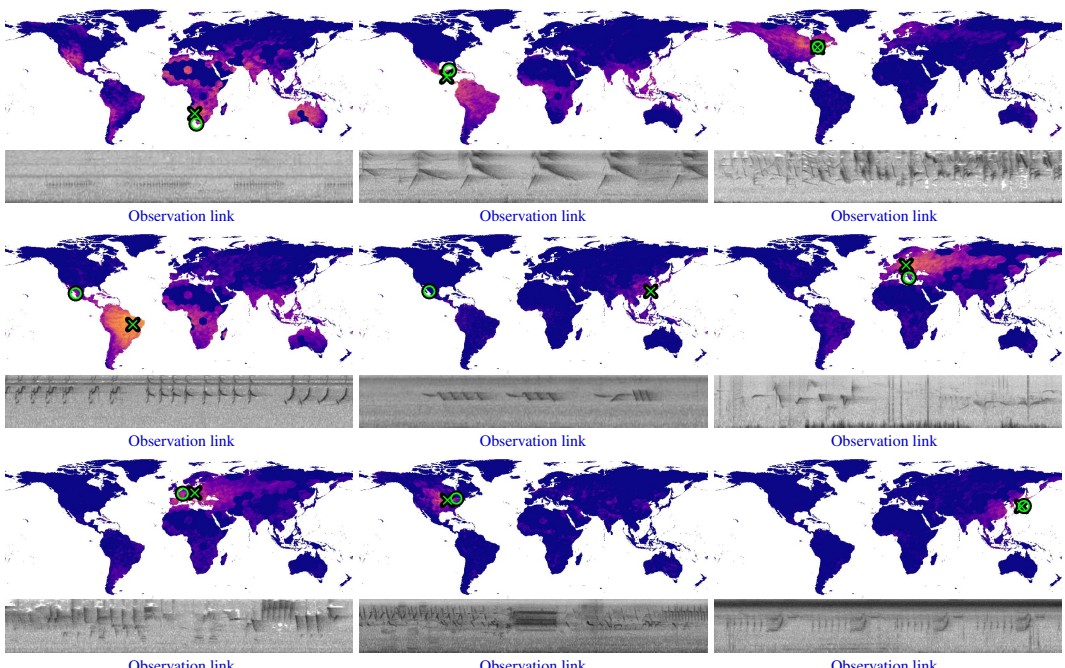

Figure A2: **Model Predictions on iNatSounds (Chasmai et al., 2024).** Heatmap shows the unnormalised likelihood of each location. Green crosses denote the final prediction (argmax) and green circles denote true location. Original observations linked below. Best viewed zoomed in.

## A.4 RETRIEVAL LOCATION GALLERIES.

Location retrieval approaches like ours require a gallery of candidate locations to retrieve from at test time. An advantage of this approach over image retrieval approaches is that candidate locations can be sampled readily (just 2 coordinates) and do not require a database at test time. However, construction of these location galleries can affect final performance. A gallery that is too sparse can lead to higher geolocation errors and a gallery that is too dense can require too much compute. See Fig A5 for visualization of different galleries.

## A.5 ADDITIONAL IMPLEMENTATION DETAILS

**Spectrogram Creation.** We take a vision approach where 1D waveforms are converted to 2D spectrograms, which can be treated as images. Following previous work (Chasmai et al., 2024), we generate spectrograms using the Short-Time Fourier transform (STFT), with a window size of 512 and a stride length of 128. Linear spaced frequencies are converted to the mel-scale (Pedersen, 1965), mapping frequencies in the range [50Hz, 11.025kHz] to 128 logarithmically spaced mel bins, better aligning with human perception of pitch change. Each audio recording is split into a set of windows of 3 seconds each, strided by 1.5 seconds. Each window is treated as an independent image, repeated thrice and resized to get a $224 \times 224 \times 3$ RGB input.

**Regression for Geolocation.** An intuitive approach is to directly regress latitude and longitude from $f_\theta(\mathbf{x})$. $h_\phi$ can be a linear layer with 2 outputs and all weights can be trained using either Euclidean distance or the differentiable Haversine distance as the loss. While the former offers simplicity, the latter is more accurate as we are computing distances on a sphere instead of a plane. The (lat, lon) coordinates are normalised to $[-1, 1]$ before being used as labels for regression. A third approach is to convert the spherical coordinates (lat, lon) to Cartesian coordinates $(x, y, z)$ and then use Euclidean distance (Perotin et al., 2019).

**Classification for Geolocation.** We can formulate geolocation as classification over a set of location bins. If the world is divided into bins, then $h_\phi$ can be trained using cross-entropy to predict the bin that contains the recording location.

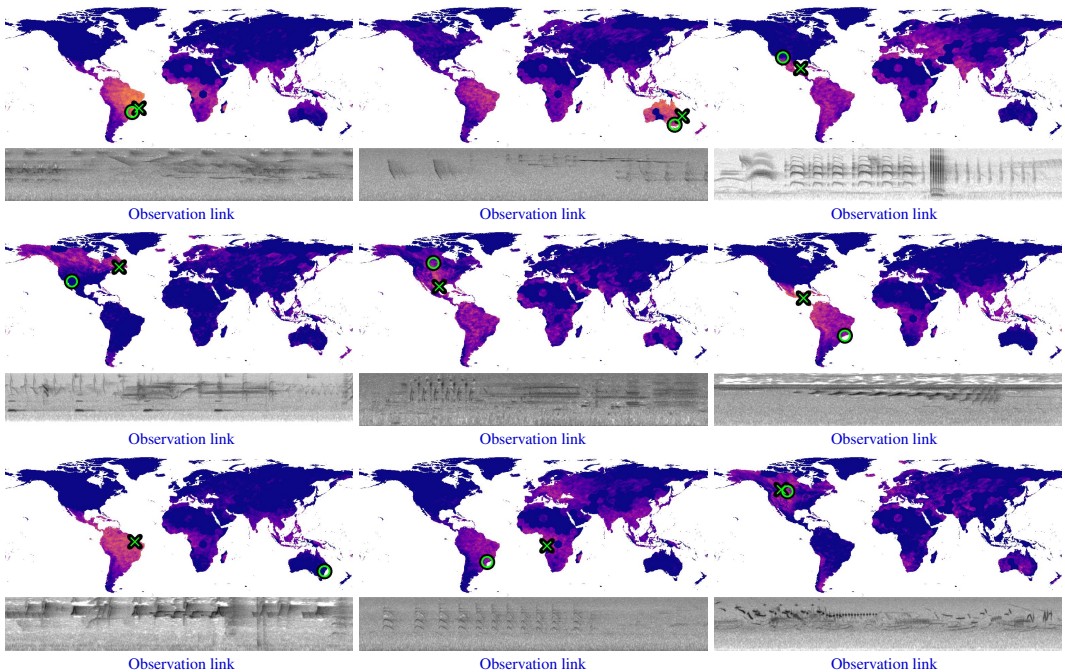

**Figure A3: Model Predictions on XCDC.** Heatmap shows the unnormalised likelihood of each location. Green crosses denote the final prediction (argmax) and green circles denote true location. Original observations linked below. Best viewed zoomed in.

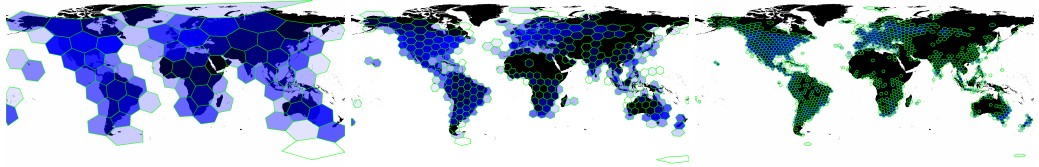

**Figure A4: Classification grids and iNatSounds Distribution.** Visualization of H3 (Brodsky, 2018) grid cells at different resolutions (0, 1 and 2 from left to right with average cell areas 436, 61 and 9 $\times 10^4 km^2$ respectively). In the hexagon itself, we show the actual distribution of the iNatSounds training set (higher opacity → more data). We use log scale for better visualization.

At test time, $h_\phi$ predicts a probability distribution over the bins and we use the center of the most likely bin as the predicted location. The bin resolution (large bin area vs small bin area) imposes constraints on the maximum achievable performance since even correctly classified (i.e., correctly binned) samples may be far from the bin center. This is likely the cause of the very poor regional performance with lowest resolution models in Table 1 (main paper). For higher resolution cells, the size of each cell is smaller, and thus, the error introduced by choosing the grid center is lower. However, higher resolution grids have more cells, or classes, which makes the classification itself somewhat harder.

We also experiment with a hierarchical approach to better handle this tradeoff. A low resolution model first predicts a large cell. Then a higher resolution model is used to predict a smaller cell within this large cell. This can be repeated for multiple levels of hierarchy. We start with resolution 0, followed by resolution 1 and finally 2. Note that this is done only at test time, so once the models at different resolution are trained, they can just be used directly for hierarchical classification.

**Model Architecture and Hyperparameters.** We show a block diagram of the AG-CLIP model architecture in Fig. A6. The pipeline has some differences at training and test times. While training, we sample windows uniformly at random from audio recordings, while at test time, we densely sample strided windows from the entire audio recording and aggregate to get recording level predictions.

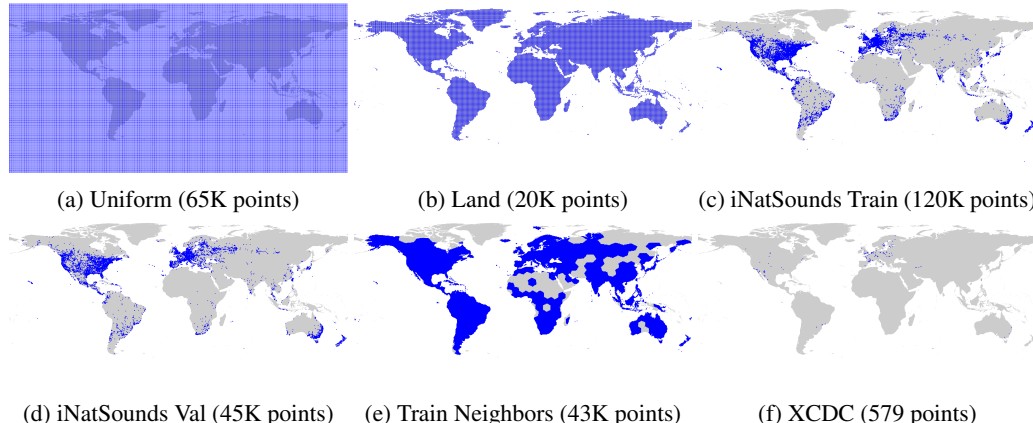

(a) Uniform (65K points)   (b) Land (20K points)   (c) iNatSounds Train (120K points)

(d) iNatSounds Val (45K points)   (e) Train Neighbors (43K points)   (f) XCDC (579 points)

Figure A5: **Location Galleries.** Visualization of different galleries for evaluating geolocation.

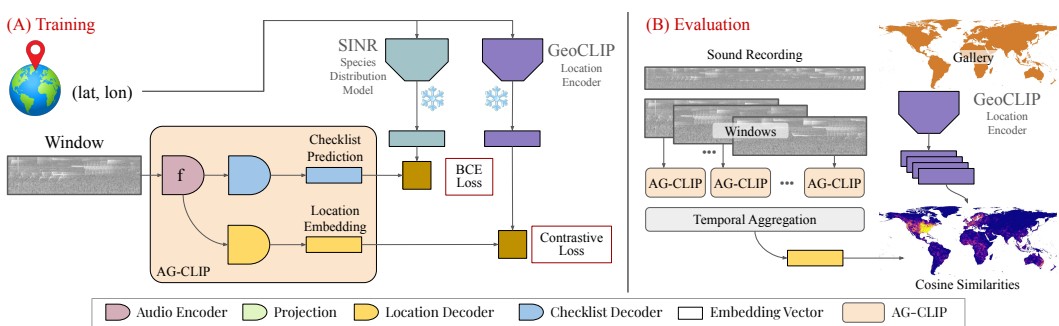

Figure A6: **Method Overview.** At training time (left), we sample a window uniformly at random from each audio recording. Features extracted by the audio encoder $f_\theta$ are fed to a checklist decoder which is used in a BCE loss with checklists given by SINR (Cole et al., 2023) as labels. Audio features $f_\theta$ are also fed to a location decoder, whose outputs are matched with corresponding Geo-CLIP (Vivanco Cepeda et al., 2024) location embeddings to get a constrastive loss. The full model is trained with the BCE loss and Contrastive loss. At test time (right), we densely sample all windows from a recording, predict final location features via the full AG-CLIP model and then aggregate these features across windows to get a recording-level prediction. Finally, we compute the similarity of this feature with GeoCLIP embeddings of a gallery of candidate locations to pick the most similar, which is the geolocation prediction.

In all experiments, we use the relatively light-weight MobileNet V3 (Howard et al., 2019) as our audio backbone to get a 1280 dimensional features from windows resized to $224 \times 224$. We choose model dimension hyperparameters to get good performance while keeping model size comparable. Our checklist prediction head learns to predict the entire species checklist given by SINR. The projection layer takes the 1280 dimensional embeddings and projects it to the checklist dimension. The checklist and location decoders are both 2 layer MLPs with hidden dimensions of 128. After these modifications, the increase in model parameters is modest, from 4.4M for GeoCLIP to 4.7M for AG-CLIP.

We initialize the audio backbones with pretrained weights for iNatSounds (Chasmai et al., 2024) species identification. We take a weighted sum of the BCE and contrastive losses, with a weight of 0.01 on the BCE loss, which reflects its magnitude relative to the contrastive loss. We train for 50 epochs with early stopping using the Nesterov accelerated SGD optimizer with a batch size of 128, weight decay of $10^{-5}$, and learning rate linearly ramped from $10^{-3}$ to $10^{-2}$ over the first 5 epochs and then cosine decayed back to $10^{-3}$ over the remaining 45 epochs. These hyperparameters were chosen with the validation set released by iNatSounds. We report test set performance.

Table A1: **Location Galleries.** Effects of the location gallery used for AG-CLIP. The same AG-CLIP model (pre-trained on iNatSounds) is given different galleries at test time to get these results.

| Gallery | # Locs | City 25km | Region 200km | Country 750km | Continent 2500km |
|---|---|---|---|---|---|
| Uniform | 65K | 01.3 | 15.0 | 38.6 | 69.2 |
| Uniform | 260K | 02.9 | 15.8 | 38.9 | 69.5 |
| Land | 20K | 01.6 | 15.4 | 38.6 | 69.6 |
| Train | 137K | 06.6 | 17.4 | 41.0 | 70.3 |
| Train Neighbors | 43K | 03.1 | 16.1 | 39.3 | 69.9 |
| Validation | 45K | 06.7 | 17.6 | 41.1 | 70.6 |
| XCDC | 576 | 00.7 | 07.4 | 28.7 | 65.7 |

**Baselines: GeoCLAP and TaxaBind.** For GeoCLAP, we get aerial imagery from the location with Google Maps API (similar to SoundingEarth (Heidler et al., 2023)), and use its image embedding as the location embedding in our retrieval setup. Instead of encoding the (lat, lon) coordinates directly with a model, we first get the corresponding aerial image, and embed this image with their image encoder. SoundingEarth, the dataset used by GeoCLAP also contains Google Maps images, and we match the zoom levels and image sizes with that dataset. We use their audio and image encoders off the shelf.

TaxaBind includes encoders for 6 different modalities, all embedding in the same shared space. We use their audio and location encoders directly, in a setup very similar to ours. Note that TaxaBind does not actually use geotagged audio. It uses geotagged images and audio-image paired data, which are used to train location-image and audio-image encoders, respectively. Since all modalities share the same space, this training strategy allows us to do audio→location retrieval, which we use for geolocation.

## A.6 TRANSFORMER TO HANDLE VARIABLE LENGTH

Models that can reason over time can potentially capture more interesting patterns. For longer recordings like those of XCDC, temporal analysis can allow a model to properly combine geographic cues from different species. Our default average pooling does this too, by way of a simple voting of different windows. We also explore if we can do better with a transformer.

We still remain in the late-fusion regime: geolocation features are predicted for each window independently and the transformer aggregates these predictions at the end. We do this with the training set so that the transformer takes in a sequence of predicted location features $N \times L$ and then predicts a single $L$-dimensional feature, where $L$ is the embedding dimension of the location encoder, which in our case is 512 for GeoCLIP. For easier batch construction, we keep the sequence length $N$ fixed at 32, padding or cropping appropriately. Note that this corresponds to around 50s of audio. We use 6 transformer layers, each with 8 heads and a hidden dimension of 128. Only the transformer layers are trained and the CNN backbone is kept frozen. We use Adam with learning rate of $10^{-3}$, weight decay of $10^{-3}$, and train for 50 epochs.

## A.7 ABLATION ON LOCATION GALLERIES

We present ablations on the location gallery in Table A1. We first experiment with galleries constructed by uniformly sampling points on the 2D world map (latitude, longitude). We sample locations in intervals of either $1°$, leading to 65K locations, or $0.5°$, leading to 260K locations. We obtain similar performances for both, with region level at 13.4% and 14.8%, respectively. This indicates that adding additional locations on a uniform grid may not improve performance much, but will lead to much higher compute. Next, we again sample uniformly ($1°$ interval), but restrict locations to only land mass. Since the earth is only about 29% land, we are left with 20K locations, but performance does not change much, and actually improves slightly for some levels (city, region, and continent). This hints at the potential of smaller, but more targeted galleries.

Table A2: **Geolocation on XCDC**. Models trained on iNatSounds training set and evaluated on XCDC.

| | Experiment | ↓ Median Error (km) | ↑ City 25km | Region 200km | Country 750km | Continent 2500km |
|---|---|---|---|---|---|---|
| Naive | Rnd Train Loc | $9915 \pm 99$ | $00.0 \pm 0.0$ | $00.1 \pm 0.2$ | $00.5 \pm 0.4$ | $03.6 \pm 0.5$ |
| Species Ranges | True Species | $449 \pm 06$ | $00.5 \pm 0.2$ | $25.8 \pm 1.4$ | $68.8 \pm 1.4$ | $99.0 \pm 0.1$ |
| | Predicted Sp (All) | $1097 \pm 18$ | $00.1 \pm 0.1$ | $02.5 \pm 0.3$ | $27.6 \pm 1.1$ | $80.0 \pm 0.3$ |
| Classify | Hierarchical | $1116 \pm 60$ | $00.0 \pm 0.0$ | $05.3 \pm 0.1$ | $31.7 \pm 1.9$ | $69.7 \pm 0.6$ |
| Retrieve | AG-CLIP (**ours**) | $1112 \pm 93$ | $00.2 \pm 0.2$ | $04.3 \pm 0.7$ | $26.3 \pm 3.5$ | $71.9 \pm 0.2$ |
| | AG-CLIP (XCDC gallery) | $912 \pm 72$ | $05.9 \pm 0.2$ | $11.6 \pm 0.7$ | $36.6 \pm 3.1$ | $73.8 \pm 0.3$ |

Table A3: **Geolocation on WABAD**. Models trained on iNatSounds training set and evaluated on WABAD (Pérez-Granados et al., 2025).

| | Experiment | ↓ Median Error (km) | ↑ City 25km | Region 200km | Country 750km | Continent 2500km |
|---|---|---|---|---|---|---|
| Regress | Euclidean | 3540 | 0.00 | 0.80 | 07.51 | 37.77 |
| | Haversine | 3853 | 0.00 | 0.86 | 08.53 | 37.66 |
| Classify | Res-0 ($430 \times 10^4 km^2$) | 4126 | 0.00 | 0.00 | 09.48 | 40.38 |
| | Res-2 ($8.6 \times 10^4 km^2$) | 5404 | 0.00 | 1.44 | 08.93 | 34.50 |
| | Hierarchical ($0 \rightarrow 1 \rightarrow 2$) | 4746 | 0.00 | 1.30 | 12.39 | 36.42 |
| Species Ranges | Annotated Species | 736 | 0.60 | 17.32 | 50.33 | 79.59 |
| | Predicted Species (Top 1) | 9678 | 0.00 | 0.37 | 01.63 | 07.23 |
| | Predicted Species (All) | 3317 | 0.35 | 2.53 | 15.69 | 44.00 |
| Retrieve | GeoCLAP | 7139 | 0.10 | 0.58 | 02.91 | 15.20 |
| | Taxabind | 7429 | 0.10 | 0.44 | 04.46 | 15.81 |
| | AG-CLIP (**ours**) | 3439 | 0.20 | 3.23 | 14.90 | 45.03 |

Next, we use the train and validation sets to construct the gallery. Keeping the locations from each geo-tagged audio in the train set, we are essentially following the distribution of the dataset itself. Regions where recordings are dense in the training set have more locations in the gallery and conversely, sparse regions in the train set are allocated fewer gallery locations. If the test location distribution is similar to the train, this would be a good gallery, well balanced between granularity and compute cost. We could use the validation set instead to construct the gallery, although the train set is a more natural choice. Compared to the land gallery, we see small improvements at the continent and country level, but much bigger improvements at the region (16.4% vs 13.8%) and city (6.5% vs 1.3%) levels. The train neighbors gallery consists of uniformly sampled points like the uniform gallery, but only from H3 (Brodsky, 2018) hexagons that contain at least one training recording. This allows denser sampling with a better control over the total number of locations. Even with 0.5° intervals, the number of locations drops to 43K, at the cost of slightly worse performance.

## A.8 AUDIO GEOLOCATION ON XCDC

We report repeated runs with mean and standard deviations for XCDC experiments in Table A2. Standard deviations are relatively low. We also include an experiment with AG-CLIP where we use the XCDC-gallery instead of iNatSounds. We see a good boost in performance, particularly at finer scales. However, these improved results are still worse that iNatSounds, supporting our hypothesis about the difficulty of our models to identify species rich data in XCDC.

## A.9 IN-THE-WILD AUDIO GEOLOCATION WITH WABAD

Our models are trained with iNatSounds, characterized by short, focal recordings. In contrast, passive acoustic monitoring (Sugai et al., 2019) (PAM) projects often collect longer recordings that capture not only target species vocalizations but also background ambient sounds. This domain shift

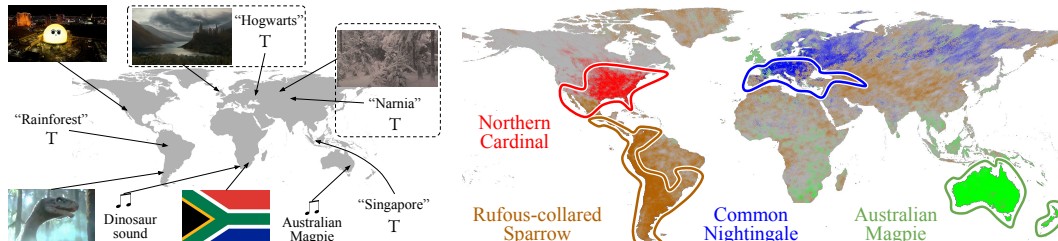

(a) **Multimodal Geolocation.** Here, we explore geolocation using audio, images, and text. We try out some fun experiments and see what the geolocation models predict for these inputs. For some of these prompts like "Narnia" or Dinosaur sounds, we do not know what the correct location should be, but it is interesting to see what the model predicts.

(b) **Soundscape Affinities of Species.** Heatmaps of cosine similarity between the average soundscape embedding of selected species and a gallery of location embeddings sampled globally. High similarity regions (darker areas) indicate locations with soundscapes that match the species' vocal features, showing alignment with known ranges and highlighting unexpected affinities with acoustically similar environments beyond typical habitats.

Figure A7: Additional experiments and visualizations

from the training distribution presents an opportunity to test our models on challenging and realistic in-the-wild settings.

WABAD (Pérez-Granados et al., 2025) is one such PAM dataset with dense annotations (timestamps and frequency ranges) covering **91K** animal vocalization from more than **1.1K** species around the world. We present the performance of our models on this dataset in Table A3. Similar to XCDC, we see significant drops compared to the performance on iNatSounds, further highlighting the challenges of this domain shift. However, the relative performance of different methods is generally consistent (lower performance of classification being an exception). AG-CLIP demonstrates better robustness than the soundscape mapping baselines Taxabind and GeoCLAP, particularly at the country and continental levels.

## A.10   ADDITIONAL DETAILS FOR MULTIMODAL GEOFORENSICS

We use Youtube clips for each movie shot we investigate. Searching for the moment where we suspect a discrepancy (based on viewer comments), we extract 10s audio clips and take a screenshot. The audio is geolocated by AG-CLIP. For the image, we use CLIP image encoder, followed by a (frozen) projection, as done by GeoCLIP. We visualize geolocation predictions of each modality of each example by arrows. These arrows show only the approximate location, but we compute the discrepancy errors exactly.

GeoCLIP keeps their image encoder frozen and trains a location encoder. This allows them to swap in CLIP's text encoder instead of image, facilitating geolocation of text. Text descriptions of visual or acoustic scenes can serve as another modality for geoforensics.

## A.11   MULTIMODAL GEOLOCATION

In AG-CLIP, we keep the location encoder frozen from GeoCLIP, who in turn had kept their image encoder frozen from CLIP. Thus, we have a set of encoders that embed audio, location, images and text to the same space, allowing us to do multimodal geolocation (see Fig A7a). Interestingly, CLIP can recognize flag images as well, and can geolocate them to the correct countries. The Las Vegas sphere (top left) finished construction after these models were trained, but they are interestingly still able to geolocate it accurately, perhaps because of some background details.

Text helps expand the scope of queries in this problem. We can ask the model to geolocate the text "rainforest", and it predicts a location close to the Amazons, the biggest rainforest in the world. Giving country names as text queries also seems to work well. As fun experiments, we also ask the model to geolocate fictional places like "Hogwarts" and "Narnia." For Narnia, the model predicts a location near Russia, which may be due to the snowy and mountainous landscapes seen in the first part of the movie. Image geolocation on a video frame from the movie also points to a similar

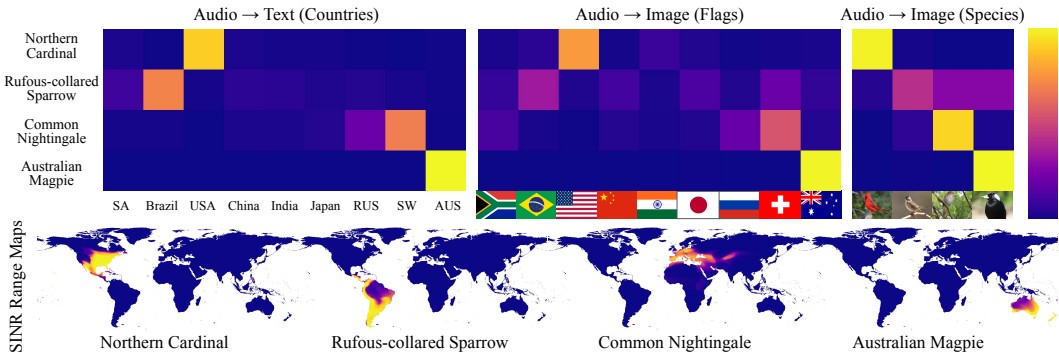

Figure A8: **Multimodal Retrieval.** We visualize some examples for Audio → Text and Audio → Image retrieval. We use the same average soundscape embeddings of species shown in Fig. A.12 and retrieve some text or image queries. We show similarity matrices for 1) Country names as text, 2) Country flags as images and 3) Same species as images. Retrieval for country texts and flag images are relatively consistent. For species image retrieval, we see some confusion for the Rufous-collared sparrow, but all species are correctly matched. At the bottom, we show the SINR range maps of these species for context of the countries these species are typically found in.

location. Similarly for Hogwarts, the predicted locations of text and image geolocation models are interestingly close. While we cannot say whether the predicted geolocations for these are correct or not, it can be a fun way to probe what these models are learning.

### A.12 SOUNDSCAPE AFFINITIES OF SPECIES.

How does the soundscape associated with a species relate to its geographic range? We explore this question using AG-CLIP and visualize the results in Fig. A7b. For each species, we compute the average audio embedding using test recordings from iNatSounds associated with that species. We then calculate the cosine similarity between this average embedding and a gallery of location embeddings uniformly sampled across the globe (restricted to land locations). The similarities are clamped to the range [0, 1] and used as the alpha channel in the heatmap visualizations.

Fig. A7b shows results for four species: Northern Cardinal, Rufous-collared Sparrow, Common Nightingale, and Australian Magpie, alongside coarse approximations of their range maps. The soundscape embedding of the Northern Cardinal aligns well with its range, but interestingly, it does not capture the western portion of its range as strongly, possibly reflecting the distinct soundscapes of eastern forests versus western deserts. For the Rufous-collared Sparrow, the soundscape feature extends far beyond its range in South America, likely due to its occupancy of semiopen habitats (i.e., villages, towns, and farmland), making it acoustically similar to many locations worldwide. The Common Nightingale's soundscape affinity stretches into Russia, suggesting a similarity in soundscapes between northern Europe and parts of Russia. Finally, the Australian Magpie's embedding aligns with its range in Australia and New Zealand but also shows unexpected affinities with deserts in Africa and rainforests in Southeast Asia, two habitats present in Australia.

These results highlight that the soundscape features learned by AG-CLIP capture both expected and surprising affinities of species soundscapes, providing insights into the relationship between species vocalizations and the broader acoustic environment.

### A.13 RETRIEVING IMAGES AND TEXT FROM AUDIO

Since we embed audio in a shared text-image-location feature space, we can use our geolocation models to perform Audio → Text and Audio → Image retrieval by simply replacing location embeddings with the corresponding modality. We present a few such experiments in Fig A8.

We use the same average soundscape embeddings of species shown in Fig. A.12. We compute similarities of these embeddings with certain text or image embeddings for retrieval. First we attempt Audio → Text by comparing each species feature with text features of country names. Notably, we

observe some confusion between Switzerland and Russia for the Common Nightingale. These similarities closely reflect the SINR (Cole et al., 2023) species range maps for corresponding species seen at the bottom of the figure. Next, we attempt Audio $\rightarrow$ Image by using images of flags of the same set of countries. We observe predictions very similar to the Text retrieval experiment, which underscores how well these modalities are paired. Finally, we repeat the Audio $\rightarrow$ Image experiment by using species images instead of flags. While there is some confusion for the Rufous-collared Sparrow, the other three species are surprisingly confident and the model retrieves all species correctly. Some multimodal retrieval tasks are often constrained by the availability of paired data from each modality. These experiments show a promising and flexible alternative of using "weakly" paired data where some but not all modalities are paired with each other.

## A.14 COMPUTE RESOURCES

We used a single A100 GPU with 80 GB GPU memory for all experiments. On a node with 1 GPU and 8 CPUs, one experiment of AG-CLIP takes 2.5 hours on iNatSounds (training + testing) and about 20 minutes for XCDC (testing only). Other approaches of species oracles, naive baselines and species ranges are much faster. Classification and regression take about the same training time as AG-CLIP, and are faster at inference time since retrieval methods need location galleries to get prediction.

## A.15 REBUTTAL TABLES AND FIGURES

For ease of reference, we include additional figures and tables requested by the reviewers here. We will move parts of these to other sections of the appendix or the main paper for the final version. Suggestions about the organisation and placement of these results are welcome!

### A.15.1 REVIEWER YZ8H.

Please see Table A4 for the amount of data and geolocation results per taxonomic class.

Table A4: **Taxonomic breakdown of iNatSounds**. Left: Number of species and recordings for different taxonomic classes. Right: Geolocation performance for each taxonomic class.

| Class | Train Species | Recordings |
|---|---|---|
| Aves | 3,846 | 111,029 |
| Insecta | 745 | 10,080 |
| Amphibia | 650 | 13,183 |
| Mammalia | 296 | 2,566 |
| Reptilia | 32 | 154 |

| Class | Median Error (km) | City 25km | Region 200km | Country 750km | Continent 2500km |
|---|---|---|---|---|---|
| Aves | 1165 | 5.9 | 15.68 | 38.79 | 69.81 |
| Insecta | 679 | 12.9 | 26.99 | 54.58 | 79.55 |
| Amphibia | 615 | 7.7 | 26.61 | 57.56 | 83.75 |
| Mammalia | 1442 | 3.6 | 11.84 | 33.27 | 61.32 |
| Reptilia | 2261 | 0.0 | 0.00 | 12.50 | 50.00 |

### A.15.2 REVIEWER ZXD9.

Please see Table A5 and Table A6 for additional experiments with other audio encoder architectures. Please see Table A7 and Fig A9 for the experiments with data resampling to mitigate the effects of species and geographic imbalance in the training data.

Table A5: **Audio Backbones.** Experiments with other backbone architectures for the audio encoder.

| Audio Backbone | Median Geolocation | City | Region | Country | Continent |
|---|---|---|---|---|---|
| MobileNet-V3 | 1082 | 6.40 | 17.2 | 41.0 | 71.2 |
| ResNet50 | 935 | 6.97 | 18.86 | 44.27 | 74.37 |
| ViT-B16 | 1069 | 7.10 | 17.79 | 41.19 | 71.24 |

Table A6: **Baselines with ResNet50.** Results for AG-CLIP and other methods for ResNet50.

| ResNet50 experiment | Median Geolocation | City | Region | Country | Continent |
|---|---|---|---|---|---|
| mse | 1650 | 0.05 | 2.67 | 23.75 | 63.05 |
| classification (res=2) | 1114 | 0.36 | 18.10 | 40.22 | 69.22 |
| Retrieval (SatCLIP) | 982 | 1.83 | 16.28 | 43.30 | 70.99 |
| Retrieval (AG-CLIP) | 935 | 6.97 | 18.86 | 44.27 | 74.37 |

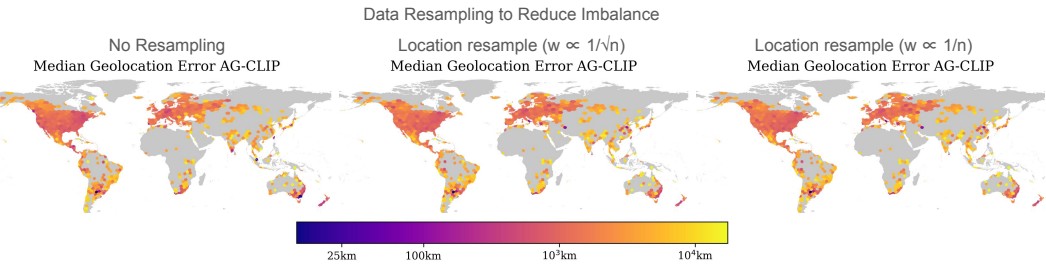

Figure A9: **Location weighted sampling.** Geographic distribution of performance for different data resampling strategies.

Table A7: **Data Resampling.** For each recording of a given species at a particular location, we compute the number of training recordings either (i) of that species or (ii) in the same h3 hexagon. With this count $n$, we assign the recording a a weight inversely proportional to $n$ ($w \propto \frac{1}{n}$) or its square root ($w \propto \frac{1}{\sqrt{n}}$). Performing a weighted random sampling with these weights allows us to balance the species and geographic distributions, respectively.

| Method | Median Error | City | Region | Country | Continent |
|---|---|---|---|---|---|
| no resampling | 1082 | 6.4 | 17.2 | 41.0 | 71.2 |
| $w \propto \frac{1}{n}$ | | | | | |
| species | 1651 | 4.3 | 12.0 | 31.2 | 59.2 |
| location | 1601 | 1.7 | 6.5 | 26.3 | 61.9 |
| $w \propto \frac{1}{\sqrt{n}}$ | | | | | |
| species | 1190 | 5.7 | 15.6 | 38.9 | 67.8 |
| location | 1204 | 4.6 | 13.6 | 37.4 | 69.2 |
| location $\times$ species | 1440 | 3.2 | 10.6 | 31.7 | 64.7 |

### A.15.3 REVIEWER QSQB

Please see Table A8 for experiments incorporating time into audio geolocation as an auxiliary training objective. Please see Table A9 and Fig A10 for experiments exploring the utility of geolocation features for species identification. Please see Fig A11 for a comparison of the geolocation error with the uncertainty of predicting SINR checklists. Please see Fig A12 for additional visualization of audio embeddings.

Table A8: **Incorporating time into audio geolocation.** Experiments with month-of-recording prediction as an auxiliary training objective. We experiment with a few different weights on the month prediction loss.

| Method | Median Error | City | Region | Country | Continent |
|---|---|---|---|---|---|
| AG-CLIP | 1082 | 06.4 | 17.2 | 41.0 | 71.2 |
| AG-CLIP + month 0.8 | 1074 | 06.2 | 16.8 | 41.2 | 71.0 |
| AG-CLIP + month 0.1 | 1113 | 06.2 | 16.6 | 40.3 | 70.5 |
| AG-CLIP + month 1.0 | 1082 | 05.8 | 16.4 | 41.0 | 71.0 |
| AG-CLIP + month 2.0 | 1122 | 05.1 | 15.0 | 40.1 | 70.5 |
| AG-CLIP + month 5.0 | 1282 | 04.3 | 12.8 | 36.4 | 67.2 |

Table A9: **Utility of features for species identification.** We explore the use of audio geolocation as a pretraining strategy for species idenficiation.

| Experiment | Top 1 | Top 5 |
|---|---|---|
| Train from scratch | 48.4 | 70.2 |
| Pretrain with Geolocation | 50.7 | 71.6 |

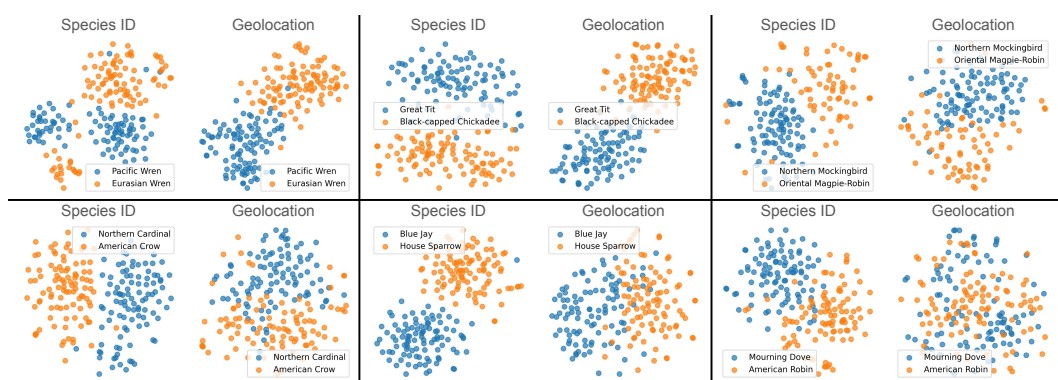

Figure A10: **Utility of features for species identification.** We visualize t-SNE projections of learned embeddings of species-ID and audio geolocation models for a few species pairs. The first row includes species pairs that tend to sound similar but are found geographically distant regions, while the second row includes species pairs that sound different but are found in the same regions.

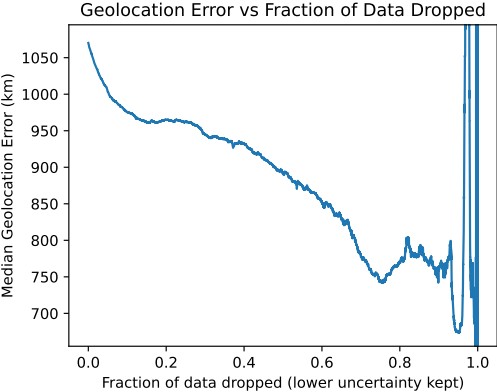

Figure A11: **Geolocation error and SINR uncertainty.** To capture the uncertainty of SINR checklists, we compute the entropy of the predicted class-wise probabilities, normalized by the number of predicted positives. We then drop the most uncertain recordings from our evaluation and plot the median geolocation errors as a function of the fraction of data dropped.

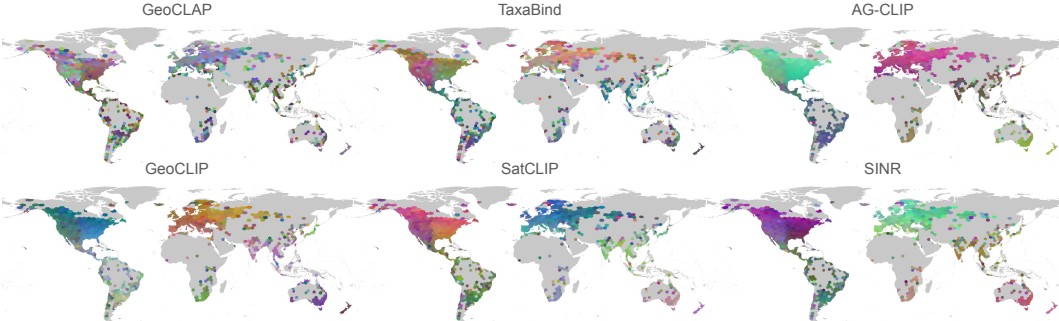

Figure A12: **Audio Embedding Visualizations.** For each h3 hexagon, we aggregate the embeddings from all test audio within that hexagon and visualize the 3D t-SNE projection of this aggregate embedding as RGB values. The colors are not comparable across different plots, but within a plot, similar colors can be interpreted as similar embeddings.

