# OpenReview forum: "Bioacoustic Geolocation: Species Sounds as Geographic Signals"
_ICLR.cc/2026/Conference — Submitted to ICLR 2026_

### Official Review · Reviewer_tt9b · 2025-10-27

**Soundness:** 2
**Presentation:** 3
**Contribution:** 1
**Rating:** 2
**Confidence:** 2

**Summary:**

The paper studies audio geolocation at global scale. It benchmarks several formulations on the iNatSounds dataset and proposes AG-CLIP, a retrieval approach that aligns audio embeddings to a GeoCLIP location space and adds an auxiliary loss to predict species-range “checklists”. The paper also introduces “species oracles”, which use predicted range maps as an upper bound for species-driven geolocation.

**Strengths:**

- The paper provides a thorough and well-structured benchmark for audio geolocation.

**Weaknesses:**

- The positioning of this work regarding the literature is not clear. This paper just describes a bunch of related lines of research (audio geolocation, image geolocation, soundscape mapping, and related audio tasks) without a clear logic. But the scientific contribution is not clear. For example, how is this work meaningfully different from image geolocation's current works? If this is just a matter of replacing an image encoder with an audio encoder (which seems to be the case), I don’t see why this is interesting for ICLR.
- AG-CLIP is essentially a straightforward retrieval model that projects audio into an existing GeoCLIP location space, plus an auxiliary multi-label loss for species checklists. Beyond the auxiliary loss, the approach is largely an adaptation of known retrieval paradigms from image geolocation. The paper itself acknowledges close links to GeoCLIP/SatCLIP/TaxaBind and frames AG-CLIP as a CLIP-style alignment with an extra head. The contribution feels incremental for ICLR.
- The core method (Sec. 3–4.1) is quite condensed; many important choices are only sketched or pushed to the appendix. For ICLR, more clarity and justification are needed in the main text.
- The paper introduces “species oracles” in the methodology section, although they are not part of the deployed method; this blurs contributions versus diagnostic analyses. As the authors note, oracles assume access to location (for checklist construction) and are not realistic at test time, so they should be clearly separated as analysis tools.
- The main novelty is empirical: framing/benchmarking and a modest extension of CLIP-style retrieval to audio with an auxiliary species loss. There is no new theory or learning principle. This feels closer to an application/benchmark paper than a core ICLR contribution.

**Questions:**

- Please specify the exact loss, normalization, temperature, and negative sampling strategy used in AG-CLIP; clarify which encoders are frozen vs. trainable and why. Also, report sensitivity to these choices.
- How are checklist targets built for each training window without leaking test-time location?

---

> ### Author Response · Authors · 2025-11-26
> **Response to Reviewer tt9b**
>
> 1. **Novelty and Contributions**
>
> We intended AG-CLIP to serve as a canonical baseline for this new task rather than a proposal for a complex new architecture. The novelty of our submission lies in the benchmark itself: we introduce the task of bioacoustic geolocation, define evaluation protocols, establish baselines across multiple modeling paradigms, and explore variations such as spatiotemporal aggregation and multimodal geoforensics. We view our work as a foundation for broader research in audio geolocation and a first step towards the exploration of bioacoustic signals for geolocation.
>
> 2. **ICLR scope**
>
> We acknowledge that our contribution is primarily empirical; however, we respectfully argue that rigorous benchmarking and novel problem formulations are critical drivers of representation learning research. Our work establishes the first standardized testbed for bioacoustic geolocation and lays the groundwork for future research in this challenging problem. We believe this submission fits well within ICLR’s subject area “applications to vision, audio, ... and other modalities”, standing alongside prior works on bioacoustics benchmarking [1, 2, 3] and applications [4, 5, 6] that have appeared at major machine learning conferences, including ICLR, NeurIPS, and ICML.
>
> 3. **Related work**
>
> We differ from related works in the task rather than the method. Audio geolocation is different and more challenging than image geolocation because cues tied to a geographic location are less salient in audio. Image geolocation relies on visual cues such as famous landmarks, landscape types, or architectural styles. However, everyday environmental or anthropogenic audio events like traffic sounds and waves are not sufficient to pinpoint location. Human speech can be a useful signal, but it is often studied in the context of accent identification instead of geolocation. Soundscape mapping is a closely related task, but with a focus on learning common patterns in sounds from a particular location rather than identifying distinctive geographic cues from a particular sound. We extend the common image geolocation paradigms to audio geolocation and are the first work to explore the potential of bioacoustic signals for global audio geolocation.
>
> We have rephrased parts of Section 2 to better articulate our relation to and distinction from related works.
>
> 4. **Species oracles in methodology**
>
> We decided to include species oracles in the methodology section because the construction of species checklists and their utility for geolocation is important to motivate the auxiliary loss in AG-CLIP. We agree about this causing potential confusion and have added a clarification to clarify it as a diagnostic tool rather than a part of the model.
>
> 5. **Condensed Methodology and Implementation Details**
>
> Given the novelty of the task, we prioritized describing the problem, the benchmark, and other experiments over detailing the specifics of our proposed approach. Please refer to model overview Fig. A6 and section A5 in the appendix for details of different encoders. Our motivation to keep the location encoders frozen was to make it easier to compare the different location encoders (Table 2b) and to facilitate emergent capabilities like multimodal retrieval (Fig. A8). For the geolocation loss, we use a CLIP-like contrastive loss with non-diagonal batch elements serving as negatives and temperature=3.6810. We use a standard binary cross entropy loss for (multiabel) checklist prediction.
>
> 6. **Checklist targets: leaking test-time location**
>
> For AG-CLIP, we construct checklists only during training. At test time, the model attempts to predict a checklist solely from the audio, without any use of test-time location.
>
> The SINR models we use to construct checklists were trained with iNaturalist observations prior to 2024 and the iNatSounds training set has recordings prior to 2023\. Since the iNatSounds test set includes recordings exclusively from 2024, this setup ensures that there is no leakage of test-time location or species information through the checklists or our training.
>
> ---
> [1] Rauch, Lukas, et al. "BirdSet: A Large-Scale Dataset for Audio Classification in Avian Bioacoustics." ICLR 2025
>
> [2] Chasmai, Mustafa, et al. "The iNaturalist sounds dataset." NeurIPS DB 2024
>
> [3] Kiskin, Ivan, et al. "HumBugDB: A Large-scale Acoustic Mosquito Dataset." NeurIPS DB 2021
>
> [4] Paradise, Orr, et al. "Towards a translative model of Sperm Whale vocalization." NeurIPS 2025
>
> [5] Boudiaf, Malik, et al. "In search for a generalizable method for source free domain adaptation." ICML 2023
>
> [6] Robinson, David, et al. "NatureLM-audio: an Audio-Language Foundation Model for Bioacoustics." ICLR 2025
>
> ---
> Thank you for the detailed feedback. We hope our responses clarify the positioning, novelty, and methodological choices of our work.

---

> > ### Comment · Reviewer_tt9b · 2025-11-27
> >
> > The authors are not answering my questions/weaknesses. I still have the same questions, some of them are:
> >
> > - What's the scientific contribution?
> > - What's the positioning of this regarding the literature?
> > - No explanation of the methodology in detail.
> >
> > Among others.
> >
> > There are some venues that are more appropriate for benchmarks, and this seems to be the main contribution of this paper.

---

> > > ### Author Response · Authors · 2025-11-28
> > >
> > > 1. **What’s the scientific contribution?**
> > >
> > > Our scientific contribution is the study of the novel task of bioacoustic geolocation. We believe that the insights from our work (such as feasibility, real-world applicability, and limitations) are valuable for both AI and ecology researchers. Our study is the first step towards numerous impactful applications of audio geolocation (such as habitat change monitoring, geoforensics, and geo-priors). Please see our response to reviewer ZXD9, point 4 for a summary of the main contributions and our response to reviewer yZ8h, point 1 for better articulation of ecological applications.
> > >
> > > 2. **What's the positioning of this regarding the literature?**
> > >
> > > Audio geolocation relies on fundamentally different cues than image-based geolocation and is inherently more challenging. While soundscape mapping, source localization, and accent identification share key methodological similarities, they do not address the unique challenges of global audio geolocation. Our work extends prior approaches and examines them in the context of the novel task of bioacoustic geolocation, laying the foundation for future exploration and potential applications.
> > >
> > > 3. **No explanation of the methodology in detail.**
> > >
> > > In our current draft, we keep high-level descriptions such as the different learning paradigms in the main paper (Sec 3, 4.1 and Table 2), and leave additional details such as architecture, loss formulations and hyperparameters to the appendix (Fig. A6 and Sec A5). We will aim to strike a better balance in the final version.

---

### Official Review · Reviewer_QsqB · 2025-10-28

**Soundness:** 4
**Presentation:** 3
**Contribution:** 2
**Rating:** 6
**Confidence:** 4

**Summary:**

The paper presents Audio-GeoCLIP, a contrastive audio-location model for geolocating species audio. The author train an audio encoder that aligns with species checklist and geolocation. In the end, the trained model can be used for predicting species checklist and geolocation of a given audio. Several experimental setups are considered for geolocating a given audio and the model performs resonably well as compared to regression, classification and retrieval-augmented approaches.

**Strengths:**

1. Overall the quality of the paper and writing is good and the motivation for the paper is meaningful.
2. Extensive experiments are conducted and a range of settings are evaluated for audio geolocation.

**Weaknesses:**

1. The paper is a very simple contrastive-based model that aligns audio with geolocation. The authors could experiment with other modalities such as text descriptions or other metadata such as time.
2. Is the trained audio encoder a good feature extractor? Can you compare how the trained audio encoder performs in other tasks such as species audio classification?
3. Can you present some analysis that compares the geolocalization error with the uncertainty of predicting the SINR checklist? Does the model perform poorly when checklists are also incorrectly predicted? Does the model need to have a good understanding of the species present in the audio to geolocalize well?
4. The authors should present some embedding visualization analysis and compare it with other models used as a baseline.
5. How do you handle the presence-only nature of audio and species observations for the task? The evaluation need to take this into consideration.

**Questions:**

Please see weaknesses.

---

> ### Author Response · Authors · 2025-11-26
> **Response to Reviewer QsqB [Part 1 of 2]**
>
> 1. **Other modalities: time**
>
> Time is an important modality for bioacoustic geolocation because of seasonal species migrations. For a simple way of integrating time into our approach, we explore using the month of the recording as an auxiliary metadata loss. We model month prediction as 12-way classification, including an additional prediction head and corresponding cross entropy loss. After a hyperparameter sweep over different loss weights, we obtain the results below (average over 3 runs):
>
> | Method | Median Error | City | Region | Country | Continent |
> | :--- | :----: | :--: | :--: | :--: | :--: |
> | AG-CLIP | 1082 | 06.4 | 17.2 | 41.0 | 71.2 |
> | AG-CLIP \+ month | 1074 | 06.2 | 16.8 | 41.2 | 71.0 |
>
> These results indicate only marginal or no improvements upon the addition of predicting month as an auxiliary task. We still believe time to be an important factor for geolocation, but leave further explorations in this direction (like season-aware range prediction models in our response to reviewer yZ8h, point 2\) for future work.
>
> 2. **Other modalities: text**
>
> iNaturalist comments and descriptions can be useful metadata for bioacoustic encoding. However, these may occasionally contain the location directly or indirectly, introducing potential leakage (if used as input) and confounding factors (if used as targets) for audio geolocation.
>
> Although we do not use text during training, we do experiment with this modality at test time. Our training strategy allows our audio encoder to indirectly be aligned with text and perform audio $\rightarrow$ text and image retrieval (Fig. A8). We observe that our audio embeddings are surprisingly well-aligned with country names (text) and flags (images), even though we do not train with these modalities.
>
> 3. **Good feature extractor for species identification**
>
> We test whether features learnt for geolocation transfer well to the task of species identification. First, we (pre)train a model for audio geolocation from scratch. Then, we (post)train this model for species identification. We compare this with a model trained for species identification from scratch, obtaining the following results:
>
> | Experiment | Top 1 | Top 5 |
> | :- | :-: | :-: |
> | Train from scratch | 48.4 | 70.2 |
> | Pretrain with Geolocation | 50.7 | 71.6 |
>
> These results suggest that the features learnt geolocation models are helpful for species identification.
>
> To further evaluate the embeddings directly, we obtain t-SNE plots of particular species pairs and compare them for embeddings learnt by geolocation and species identification models. Please see these plots in Fig. A10 in the appendix-rebuttal. These plots suggest that geolocation models are:
>
> - Good at differentiating species that sound similar but are from different geographic regions
> - Bad at differentiating between species that sound distinct but are from similar geographic regions
>
> Overall, we believe that geolocation models learn features that can aid in species identification, but are not sufficient on their own. Incorporating geolocation as an auxiliary task or a pretraining strategy could be effective ways to combine the two.
>
> 4. **Geolocation error and SINR uncertainty**
>
> Great question. We visualize this trend in Fig. A11 in appendix-rebuttal. To quantify the uncertainty of SINR checklists, we compute the entropy of the predicted class-wise probabilities, normalized by the number of predicted positives. We then iteratively drop the most uncertain recordings (with highest entropy) from our evaluation and plot the median geolocation error as a function of the fraction of data removed.
>
> The plot indicates that geolocation performance tends to improve with lower uncertainty in SINR checklist predictions. For the full dataset, the median geolocation error is 1082 km, which drops to around 760 km for the top 10% of recordings with the lowest SINR uncertainty. The plot is quite noisy towards the right end (very small fractions of data left), reflecting exceptions to this trend with poor geolocation performance despite highly confident SINR predictions. Broadly, lower uncertainty in the SINR checklists likely corresponds to regions with more distinctive species compositions and this trend suggests that such regions are easier to geolocate.

---

> ### Author Response · Authors · 2025-11-26
> **Response to Reviewer QsqB [Part 2 of 2]**
>
> 5. **Embedding visualizations**
>
> We visualize the learned embeddings with t-SNE plots for a few species pairs to understand their utility for species identification in Fig. A10.
>
> We also visualize the learned embeddings for audio from different parts of the world in Fig. A12. For each h3 hexagon, we aggregate the embeddings from all test audio within that hexagon and visualize the 3D t-SNE projection of this aggregate embedding as RGB values. The colors are not comparable across different plots, but within a plot, similar colors can be interpreted as similar embeddings.
>
> For the baselines Taxabind and GeoCLAP, the embeddings seem somewhat noisy, sometimes having significantly different embeddings for neighboring regions. For Taxabind, we see some consistency for the eastern US. Our audio embeddings, with SatCLIP, SINR or GeoCLIP location encoders seem much more uniform, with noticeable global patterns.
>
> 6. **Presence-only audio and species observations**
>
> The lack of species negatives in each audio recording should not be a factor in our evaluation setup since our ground truth is simply the location of each recording (not using species labels). It does affect some of our baselines such as the “Species Ranges - Annotated Species”, where we look at the range maps for annotated species for geolocation. Here, we assume that all non-annotated species are indeed negatives. The use of predicted species yielding better results than annotated species for “Species Ranges” (Table 1\) is likely an artifact of this presence-only nature, where the species-ID model may be picking up some species that were incorrectly assumed as negative.
>
> ---
> We appreciate your insightful comments and suggestions. We hope our responses and additional analyses address your concerns and clarify the strengths and limitations of our approach.

---

### Official Review · Reviewer_ZXD9 · 2025-10-31

**Soundness:** 2
**Presentation:** 2
**Contribution:** 1
**Rating:** 2
**Confidence:** 3

**Summary:**

The paper introduces AG-CLIP, a model that integrates audio signals from species with geolocation information. In addition, the authors propose a new task that uses acoustic signals for geolocation prediction. The benchmark is evaluated on the iNatSounds dataset.

**Strengths:**

* The paper presents an extensive set of experiments that explore the potential of audio signals for geolocation tasks.
* The authors show that an acoustic signal (i.e., a wildlife audio dataset) can be an important modality for geolocation tasks

**Weaknesses:**

* The audio encoder used in AG-CLIP is MobileNet-V3, which is not specifically designed for audio processing. More suitable models, such as EsResNet or AudioCLIP, which have been trained on audio datasets like UrbanSound8K, could likely improve performance.
* The proposed AG-CLIP model, one of the paper’s main contributions, is only briefly described in the main text, making it difficult to fully assess the novelty or implementation details.
* The retrieval evaluation may not be directly comparable: AG-CLIP is trained on the full dataset iNatSounds, whereas GeoCLAP and TaxaBind are trained on different datasets. This discrepancy raises concerns about the fairness of the comparison. A stronger baseline is to fine-tune GeoCLAP and Taxabind on the same audio dataset.
* The novelty of the work appears limited. The proposed method largely reuses existing components for audio encoding, and the results do not provide sufficient insight into the model’s limitations or reasons for its weaker performance on certain tasks.

**Questions:**

* How was the class imbalance in the dataset handled? Was any resampling or balancing strategy applied?
* What factors explain the model’s low accuracy for the City and Region categories in Table 1? Are there specific limitations or failure cases responsible for this performance gap?

---

> ### Author Response · Authors · 2025-11-26
> **Response to Reviewer ZXD9 [Part 1 of 2]**
>
> We were surprised by some of the weaknesses raised in the reviews, which suggests that parts of our presentation may not have been sufficiently clear. We hope the responses below help clarify these concerns.
>
> 1. **Other audio encoders**
>
> Despite the reviewer’s concerns, converting audio to spectrograms and applying image-based deep classifiers is a standard and widely adopted approach in bioacoustics and general audio processing. Our choice of MobileNet-V3 was motivated by its strong performance-to-resource requirement ratio, which allowed us to iterate quickly through design choices. That said, our experiments do evaluate a broad range of architectures, including those comparable to the ones highlighted by the reviewer.
>
> We explore Wav2CLIP and CLAP pretrained encoders as alternative audio encoders in Table 2 (a). Our setup is very similar to EsResNet: we also extract 2D spectrograms and treat them as images for vision encoders. AG-CLIP results with ResNet50 and ViT-B16 as the vision backbones instead of MobileNet are here:
>
> | Audio Backbone | Median Geolocation | City | Region | Country | Continent |
> | - | -: | - | - | - | - |
> | MobileNet-V3 | 1082 | 6.40 | 17.2 | 41.0 | 71.2 |
> | ResNet50 | 935 | 6.97 | 18.9 | 44.3 | 74.4 |
> | ViT-B16 | 1069 | 7.10 | 17.8 | 41.2 | 71.2 |
>
> We repeat a few other baselines with the ResNet 50 backbone in the table below. The trends between different geolocation paradigms are very similar to what we observed for MobileNet in Table 1\.
>
> | ResNet50 experiment | Median Geolocation | City | Region | Country | Continent |
> | - | -: | - | - | - | - |
> Regression | 1650 | 0.05 | 02.7 | 23.8 | 63.0
> Classification (res=2) | 1114 | 0.36 | 18.1 | 40.2 | 69.2
> Retrieval (SatCLIP) | 982 | 1.83 | 16.3 | 43.3 | 71.0
> Retrieval (AG-CLIP) | 935 | 6.97 | 18.9 | 44.3 | 74.4
>
> 2. **AG-CLIP methodological details**
>
> This choice was primarily motivated by space constraints. Given the novelty of the task, we prioritized describing the problem, the benchmark, and other experiments over detailing the specifics of our proposed approach. These details are, however, included in the supplementary material (see model overview in Fig. A6 and Section A5). We believe that our methodology section includes sufficient context for a reader to follow the experiments and discussions. In the revision, we will aim to strike a better balance between describing the task and presenting architectural details.
>
> 3. **GeoCLAP and TaxaBind fair comparisons**
>
> Direct comparison is challenging due to the differences in architecture. However, our main goal in comparing with GeoCLAP and TaxaBind was to understand the influence of different training paradigms:
>
> - GeoCLAP: train on urban sounds, satellite images as location-proxy
> - TaxaBind: train on species sounds, implicit audio-location pairing via a third modality
> - Ours: train on species sounds, explicit audio-location pairing
>
> TaxaBind is trained on a larger, distinct iNaturalist-sourced dataset, so domain shift is less likely to be a factor. GeoCLAP and TaxaBind use stronger transformer-based backbones, which could be an advantage over our MobileNet based models. Outperforming these models with a smaller backbone strengthens our claims regarding the benefits of bioacoustic training (GeoCLAP) and explicit audio-location pairing (TaxaBind).
>
> For stronger retrieval baselines, please refer to Table 2 (b). Here, we train audio encoders with different (frozen) location encoders. The location encoder models themselves were trained for other geospatial tasks like image geolocation (GeoCLIP), satellite imagery (SatCLIP), and species range estimation (SINR).
>
> 4. **Novelty and Insights**
>
> We study audio geolocation in natural domains, a novel task with numerous practical applications, as highlighted in the paper (and in our response to reviews yZ8h). Our main findings are: (1) audio geolocation is feasible to a meaningful extent (we can localize within \~750 km with 41% accuracy), (2) models transfer well to real-world settings, but species overlap and difficulty in identification limit performance, (3) aggregating information across space and time, common in applications like iNaturalist and Merlin, substantially improves performance, and (4) the task remains significantly more challenging than image-based geolocation. We believe these insights are valuable for both AI and ecology researchers. We view our work as a foundation for broader research in audio geolocation and a first step towards the exploration of bioacoustic signals in this problem.

---

> ### Author Response · Authors · 2025-11-26
> **Response to Reviewer ZXD9 [Part 2 of 2]**
>
> 5. **Class imbalance**
>
> The class balance, both in terms of species and locations, is an important limitation for this work. The train/val/test splits in iNatSounds were constructed to mitigate these imbalances to some extent. The number of audio recordings per species is capped at 1000 for the training set. This sub-sampling is done as follows: first they are clustered geographically and then recordings are sampled at random from clusters in a round-robin fashion to increase geographic diversity. Despite this, there is still geographical bias in our model (Fig 3, right).
>
> We additionally explore resampling strategies to mitigate this species and geographic imbalance. For each recording of a given species at a particular location, we compute the number of training recordings either (i) of that species or (ii) in the same h3 hexagon. With this count $n$, we assign the recording a a weight inversely proportional to $n$ ($w \propto \frac{1}{n}$) or its square root ($w \propto \frac{1}{\sqrt{n}}$). Performing a weighted random sampling with these weights allows us to balance the species and geographic distributions, respectively. Our results are presented below:
>
> | Method | Median Error | City | Region | Country | Continent |
> | :- | -: | - | - | - | - |
> | no resampling | 1082 | 6.4 | 17.2 | 41.0 | 71.2 |
> | |
> | $w \propto \frac{1}{n}$ |
> | species | 1651 | 4.3 | 12.0 | 31.2 | 59.2 |
> | location | 1601 | 1.7 | 6.5 | 26.3 | 61.9 |
> ||
> | $w \propto \frac{1}{\sqrt{n}}$ |
> | species | 1190 | 5.7 | 15.6 | 38.9 | 67.8 |
> | location | 1204 | 4.6 | 13.6 | 37.4 | 69.2 |
> | location $\times$ species | 1440 | 3.2 | 10.6 | 31.7 | 64.7 |
>
> We see worse overall performance for each of these resampling strategies, with significant drops in region performance. Location based resampling, with a weight inversely proportional to the square root of the count performs best amongst the different reweighting strategies. We also look at the geographic distribution of performance for these strategies in Fig A9 (appendix-rebuttal). We see improvements in a few regions in east Europe and east Asia, but changes in the global distribution of performance are not that obvious.
>
>
> 6. **Low City and Region Performance**
>
> We suspect that audio geolocation is just that challenging\!
>
> A recent image geolocation model [1] achieves 9% to 40% accuracy at the city level for different benchmarks. Our best city-level performance is 6.4%. We argue that audio geolocation is harder than its image counterpart. Many of our short audio recordings simply do not contain enough information to geolocate to within 25km (for city performance). Our species oracles suggest a 10.0% city performance can be achieved if there is information of 10 distinct species in each recording, a conservative estimate of the amount of information we could expect in the average 20s recording in our dataset. Challenges in identifying these species in real-world settings is another factor affecting geolocation performance. Spatiotemporal aggregation (Table 4\) and stronger species identification are promising ways to improve geolocation performance.
>
> [1] Haas, Lukas, et al. "Pigeon: Predicting image geolocations." Proceedings of the IEEE/CVF Conference on Computer Vision and Pattern Recognition. 2024\.
>
> ---
>
> We appreciate your comments and hope our response clarifies our novelty and contributions, inherent difficulty of the task, the effects of architectural choices, and our baseline comparisons.

---

### Official Review · Reviewer_yZ8h · 2025-11-06

**Soundness:** 2
**Presentation:** 2
**Contribution:** 2
**Rating:** 4
**Confidence:** 4

**Summary:**

This paper presents the first large-scale study on bioacoustic geolocation, investigating whether the geographic location of a recording can be inferred solely from audio, particularly wildlife sounds. Using the iNatSounds and XCDC datasets, the authors benchmark multiple geolocation strategies, introduce AG-CLIP (a hybrid retrieval model combining audio embeddings with species range predictions), and demonstrate improved performance through spatiotemporal aggregation. They also explore multimodal geo-forensics, showcasing modality mismatches in movie soundtracks.

**Strengths:**

The paper’s primary strength is its originality: it represents the first attempt to scale bioacoustic geolocation globally, integrating species range predictions into a deep learning framework. Methodologically, it is thorough, with multiple model formulations, ablations, and dataset comparisons, including a valuable new benchmark of species-rich recordings (XCDC). The interdisciplinary relevance is high, as it bridges machine learning and ecology by operationalizing expert ecological reasoning into a scalable computational model. The scalability and extensibility of the approach suggest broad applicability in areas such as biodiversity assessment, detecting environmental change, and analyzing geospatial multimedia.

**Weaknesses:**

Despite its strengths, the study lacks detail in ecological interpretation. The manuscript does not clearly articulate how this method may benefit biodiversity monitoring, conservation decision-making, or citizen science, which may limit its relevance for ecological practitioners. Temporal factors, such as seasonality or animal migration, are not accounted for, raising concerns about the generalizability of the results in real-world settings. The performance gap between curated datasets (such as iNatSounds) and passive, multi-species recordings (XCDC) underscores the need for more robust multi-species classifiers or domain adaptation. Finally, the taxonomic composition of the dataset remains unclear, making it difficult to determine how representative or generalizable the model is across different animal groups.

**Questions:**

Which taxa dominate the 5,500 species in the iNatSounds dataset? A breakdown would help clarify whether the model is primarily learning from bird vocalizations or adapting more broadly to less-studied groups, such as amphibians or mammals, which vocalize less frequently or in different frequency bands.

How is seasonal variability handled in species range predictions (e.g., via SINR)? Many species migrate or vocalize only at certain times of the year, so range-based inference may introduce errors if not seasonally aware. If this were not considered, what are the expected effects on performance, especially in temperate or migratory-dominant ecosystems?

Can the authors expand on the factors contributing to the model’s lower performance on the XCDC dataset? Is it due to increased species overlap, background noise, distribution shift from curated to passive data, or other issues? Insights into this failure mode would be valuable for future work on multi-species bioacoustics.

What real-world ecological applications do the authors envision for this approach beyond academic benchmarking? For example, could this aid in monitoring habitat changes, tracking invasive species, or informing conservation planning in data-poor regions?

Have the authors considered incorporating anthropogenic or environmental audio cues (e.g., traffic, river flow) as additional signals for geolocation to complement bioacoustic species detection in urban or non-wildlife-dominant settings?

---

> ### Author Response · Authors · 2025-11-26
> **Response to Reviewer yZ8h [Part 1 of 2]**
>
> 1. **Ecological Applications**
>
> Monitoring habitat change is a particularly promising real-world application of audio geolocation. Unannotated soundscapes inherently capture aspects of a habitat, and shifts in predicted locations over time can indicate ecological change. For example, if a particular region undergoes deforestation or urbanization, geolocation predictions would drift from the original forested region to neighboring urban regions. After events such as forest fires, predictions may shift towards deserts and arid regions. Longer term changes such as global warming could cause northern regions to become (and sound) more similar to equatorial regions. Crucially, this type of monitoring does not require expensive species annotations, which is often a bottleneck in bioacoustics work.
>
> For tracking invasive species or informing conservation planning in data-poor regions, we expect species identification or species range estimation models to remain more suitable. However, audio geolocation could provide complementary habitat context in settings where annotation is limited.
>
> Another practical application of geolocation is its ability to improve species prediction with geographical context. Apps such as Merlin Sound ID explicitly rely on location to constrain the set of plausible species and reduce confusions between acoustically similar species. However, the location may not be available for many audio recordings collected in areas without cell coverage or GPS availability. In such cases, our approach can recover this ‘geographical context’ by aggregating information across audio files, enabling downstream models to implicitly benefit from location cues even when they are not provided.
>
> More broadly, just as image geolocation opened the door to a wide range of use cases, we believe audio geolocation is an emerging capability that will find its place as tools and datasets mature. The scientific challenge alone is also compelling: can we localize a place based solely on what it sounds like? We see our work as laying the groundwork for future efforts and as a first step towards ever more diverse and ambitious audio geolocation applications.
>
> 2. **Seasonality in SINR species ranges**
>
> The short answer is that we do not currently model seasonal variability, which is indeed a limitation of our approach. Beyond birds (e.g., via eBird Status & Trends), there is limited spatiotemporal data (particularly on platforms like iNaturalist) to support learning a full spatiotemporal SINR model. Given the challenges of species distribution modeling in general (long-tail of species observations, sparse, presence-only data, sampling bias, etc.), it has been more effective to train SINR models using observations aggregated over time. As these biodiversity platforms continue to grow, it might be feasible to develop spatiotemporal models for a wide range of species in the future.
>
> 3. **Lower performance in XCDC and WABAD**
>
> For XCDC, we suspect that our models only identify a fraction of the species in each recording. This is further corroborated by similar performance for the points on the lower side of Table 3-right. We believe that increased species overlap, and consequently, increased difficulty of species identification is the most likely culprit. This finding is reinforced by our spatiotemporal aggregation experiments, where recordings with higher species diversity but low species overlap yield better results.
>
> WABAD is a passive acoustic dataset. In addition to more species overlap, the recordings here also have significant background noise and a domain shift from the focal recordings in the training set. Besides, PAM datasets are often recorded away from human population centers and the end-to-end models may not have anthropogenic or environmental signals like airplane or wave sounds to refine geolocation estimates.
>
> Lastly, another contributing factor to consider is the geographical skew in the iNatSounds test set. We may be seeing somewhat inflated performance because there are more recordings from North American and Western European regions, where we tend to see better performance already (Fig 3-right). XCDC and WABAD also have more recordings from these regions, but the skew is not as pronounced.

---

> ### Author Response · Authors · 2025-11-26
> **Response to Reviewer yZ8h [Part 2 of 2]**
>
> 4. **Taxonomic breakdown**
>
> We include the taxonomic breakdown of the iNatSounds [1] training set for easier reference. Please refer to their paper for more details.
>
>  Class | Train | Train
> | - | - | -
> | | Species | Recordings
> Aves| 3,846 | 111,029
> Insecta| 745 | 10,080
> Amphibia| 650| 13,183
> Mammalia|296| 2,566
> Reptilia| 32 | 154
> |
> Total| 5,569 | 137,012
>
> It is dominated by birds in both the number of species and recordings. Amphibians and Mammals make up only 11.7% and 5.3% of the species, respectively.
>
> We present the geolocation performance for these taxonomic groups below:
>
> | Class | Median Geolocation | City | Region | Country | Continent |
> | - | - | - | - | - | - |
> | Aves | 1165 | 05.9 | 15.7 | 38.8 | 69.8 |
> | Insecta | 679 | 12.9 | 27.0 | 54.6 | 79.6 |
> | Amphibia | 615 | 07.7 | 26.6 | 57.6 | 83.8 |
> | Mammalia | 1442 | 03.6 | 11.8 | 33.3 | 61.3 |
> | Reptilia | 2261 | 00.0 | 00.0 | 12.5 | 50.0 |
>
> Despite fewer training recordings for insects and amphibians, their geolocation performance is better than that for birds. This suggests that insects and amphibians are inherently easier to geolocate than birds. Other factors like geographic and time-of-day shifts (eg: night time for insects, day for birds) may affect this performance as well.
>
> 5. **Anthropogenic and Environmental audio cues**
>
> In our experiments, we observe that the end-to-end (audio $\rightarrow$ location) model AG-CLIP performs better than species-only methods (audio $\rightarrow$ species $\rightarrow$ location), even with use of ground truth species (Table 1, Species ranges vs AG-CLIP). This is likely due to the model’s ability to capture anthropogenic and environmental sounds in iNatSounds recordings. However, we suspect that these sounds are not sufficiently specific, and serve as additional clues instead of primary signals. Extending our study to anthropogenic and environmental sounds is out of scope for this paper, but is a natural next step for future work.
>
> [1] Chasmai, Mustafa, et al. "The iNaturalist sounds dataset." *Advances in Neural Information Processing Systems* 37 (2024): 132524-132544.
>
> ---
>
> We appreciate your perspective and hope our response clarifies the ecological interpretation, applicability across different animal groups and in real-world settings, and potential impact of our work. We hope that our work can invite further research into this challenging problem and motivate other interesting applications of audio geolocation.

---

### Author Response · Authors · 2025-11-26
**General comments to the reviewers**

We thank the reviewers for their thoughtful and constructive feedback. We appreciate their recognition of the motivation and originality of the bioacoustic geolocation problem (yZ8h, tt9b, QsqB); the thoroughness of our experimental setup (yZ8h, ZXD9, QsqB); the quality of our writing (QsqB); and the potential applications and impact (yZ8h). We are pleased that the reviewers found value in the species-rich audio geolocation benchmark XCDC (yZ8h).

The reviewers’ primary concerns and questions relate to our novelty and contributions, better articulation of potential applications, additional analyses of audio embeddings, and addressing key limitations of current approaches, including domain shift, seasonality and class imbalance. We address specific concerns raised by reviewers as separate responses.

Changes within the main manuscript are highlighted in blue. For ease of reference, we include additional figures and tables requested by the reviewers in Sec. A15 of the appendix. We will move parts of these to other sections of the appendix or the main paper for the final version. Suggestions about the organisation and placement of these results are welcome!

---

### Meta-Review · Area_Chair_QNzD · 2025-12-28

**Summary:**

While the task of bioacoustic geolocation is undoubtedly interesting and carries potential value for biodiversity monitoring, I have decided to recommend a rejection for this submission due to limited technical novelty and an insufficient exploration of its real-world ecological impact. Reviewer yZ8h noted a significant lack of detail regarding ecological interpretation and how the method would practically benefit conservation practitioners. Reviewer ZXD9 argued that the technical novelty is limited as the method largely reuses existing components with standard encoders. Reviewer QsqB pointed out that the model is a relatively simple contrastive framework that lacks a deep analysis of uncertainty or multi-modal integration during training. Finally, Reviewer tt9b expressed strong concerns that the work feels more like an application or benchmark paper than a core ICLR contribution, lacking new learning principles.

**Reviewer Concerns:**

The authors provided a detailed rebuttal that addressed several technical and data-centric queries, yet core conceptual concerns remain unsolved:

Addressed:

+ Taxonomic Clarity: The authors provided a helpful taxonomic breakdown of the iNatSounds dataset and performance metrics across different animal groups.
+ Class Imbalance: The rebuttal clarified the sampling strategies used to mitigate geographic bias and explored additional resampling techniques.
+Feature Extraction: The authors demonstrated that the geolocation-trained features can aid in species identification, showing some transferability.

Outstanding:

- Scientific Contribution: Reviewer tt9b's core concern remains that the contribution is a modest empirical extension of CLIP-style retrieval and lacks a fundamental technical breakthrough.
- Technical Novelty: Reviewer ZXD9’s point about the simplicity of the architecture and the limited insights into failure modes was not fully resolved by the provided baseline updates.
- Ecological Impact: While the authors discussed potential applications like monitoring habitat shifts after forest fires, the manuscript still lacks the extensive, grounded discussion on how these models would be operationalized by ecological practitioners or how they handle fundamental issues like seasonality, especially when the technique part is not significant.

**Reviewer Scores:**

Based on the above issues, I think Reviewer yZ8h, ZXD9, and tt9b will remain at their score, and QsqB may lower his/her score as the limited merits after introducing more modalities.

---

### Decision · Program_Chairs · 2026-01-26

Reject